# Combinatorial treatment with traditional medicinal preparations and VEGFR-tyrosine kinase inhibitors for middle-advanced primary liver cancer: A systematic review and meta-analysis

**Hui-Bo Yu**[1,2‡], **Jia-Qi Hu**[1,2‡], **Bao-Jin Han**[1,2‡], **Yan-Yuan Du**[1], **Shun-Tai Chen**[2], **Xin Chen**[1], **Hong-Tai Xiong**[1], **Jin Gao**[1,2], **Hong-Gang Zheng**[1] *

**1** Department of Oncology, Guang'anmen Hospital, China Academy of Chinese Medical Sciences, Beijing, China, **2** Graduate College, Beijing University of Chinese Medicine, Beijing, China

‡ HBY, JQH and BJH are share co-first author and contributed equally to this work.
* honggangzheng@126.com

## Abstract

### Background

This study aimed to investigate the therapeutic efficacy and safety of Traditional medicine preparations (TMPs) given in combination with vascular endothelial growth factor receptor (VEGFR)-associated multi-targeted tyrosine kinase inhibitors (TKIs) for the treatment of middle to advanced-stage primary liver cancer (PLC).

### Methods

This systematic literature survey employed 10 electronic databases and 2 clinical trial registration platforms to identify relevant studies on the use of TMPs + VEGFR-TKIs to treat patients with middle-advanced PLC. Furthermore, a meta-analysis was performed following the PRISMA guidelines using the risk ratio (RR) at 95% confidence intervals (CI) or standardized mean difference as effect measures.

### Results

A total of 26 studies comprising 1678 middle-advanced PLC patients were selected. The meta-analysis revealed that compared with VEGFR-TKI mono-treatment, the co-therapy of TMPs + VEGFR-TKIs considerably enhanced the objective response rate (RR = 1.49, 95% CI: 1.31–1.69), disease control rate (RR = 1.23, 95% CI: 1.16–1.30), and one-year overall survival (RR = 1.49, 95% CI: 1.28–1.74). Furthermore, the co-therapy was associated with reduced incidences of liver dysfunction (RR = 0.64, 95% CI: 0.45–0.91), proteinuria (RR = 0.43, 95% CI: 0.24–0.75), hypertension (RR = 0.66, 95% CI: 0.53–0.83), hand-foot skin reactions (RR = 0.63, 95% CI: 0.49–0.80), myelosuppression (RR = 0.63, 95% CI: 0.46–

**Funding:** This study was supported by the Scientific and Technological Innovation Project of the China Academy of Chinese Medical Sciences (No. CI2021A01804), the National Natural Science Foundation of China (Grant No. 82174463), and the Innovation Team and Talents Cultivation Program of the National Administration of Traditional Chinese Medicine (No. ZYYCXTD-C-202205). The funders had no role in study design, data collection and analysis, decision to publish, or preparation of the manuscript. No additional external funding was received for this study.

**Competing interests:** The authors have declared that no competing interests exist.

**Abbreviations:** ADRs, adverse drug reactions; AFP, alpha fetoprotein; ALT, alanine aminotransferase; CFDA, China Food and Drug Administration; CI, confidence intervals; CR, complete response; CRC, colorectal cancer cells; CSCO, Chinese Society of Clinical Oncology; DCR, disease control rate; FDA, Food and Drug Administration; FEM, fixed-effects model; GRADE, Grading of Recommendations Assessment Development and Evaluation criteria; HCC, Hepatocellular Carcinoma; IPC, intraperitoneal chemotherapy; KPS, Karnofsky Performance Status; MA, malignant ascites; MD, mean difference; MDR, multidrug resistance; MT, matrine; MVD, micro vessel density; NCI-CTCAE, the national Cancer Institute-Common Toxicity Criteria for Adverse Events; ORR, objective response rate; OS, overall survival; PC, pancreatic cancer; PD, progressive disease; PLC, Primary liver cancer; PR, partial response; QoL, quality of life; RCTs, randomized controlled trials; REM, random-effects model; RR, risk ratio; SD, stable disease; SM, statistical method; SMD, standardized mean difference; TACE, transcatheter arterial chemoembolization; TKIs, targeted tyrosine kinase inhibitors; TMPs, Traditional medicine preparations; VEGFR, vascular endothelial growth factor receptor.

0.87), and gastrointestinal reactions (RR = 0.64, 95% CI: 0.45–0.92). Moreover, the co-therapy indicated no increase in the incidences of rash and fatigue.

## Conclusion

This systematic analysis revealed that co-therapy with TMPs + VEGFR-TKIs has a higher effectiveness and safety profile for treating middle-advanced PLC patients. However, further validation using randomized control trials is required.

## PROSPERO registration no

CRD42022350634.

## Introduction

Malignant transformation of hepatocytes and bile duct cells leads to the development of primary liver cancer (PLC). It is among the most prevalent primary tumor manifestations. In 2020, PLC was ranked 6th most prevalent malignancy and the 3rd highest cause of death globally, accounting for about 910,000 novel PLC patients and 830,000 PLC-related deaths [1]. Currently, surgery is the most effective method for achieving long-term survival in PLC patients. However, because of the lack of apparent symptoms at early stages, PLC onset is relatively insidious, and its progression is rapid. Therefore, most PLC patients are diagnosed at intermediate or advanced stages when the best opportunity for radical treatment has been lost.

The Vascular Endothelial Growth Factor (VEGF) family comprises VEGF-A, VEGF-B, VEGF-C, VEGF-D, and placental growth factor [2]. These proteins predominantly bind VEGF receptor-1 (VEGFR-1) or Fms-like Tyrosine Kinase-1, and VEGFR-2 (also called kinase insert domain-containing receptor) [2, 3]. This interaction activates downstream signaling pathways crucial for endothelial cell proliferation, differentiation, and migration, as well as the regulation of vascular permeability, which are essential for angiogenesis. Angiogenesis ensures a steady supply of oxygen and nutrients during tumor development and progression. VEGF, secreted by tumor cells and their microenvironment, binds to VEGFR-2 and exerts a pivotal role in vascular permeability and neo-angiogenesis [4]. In 1993, a monoclonal antibody targeting and neutralizing VEGFA was identified to inhibit tumor growth in xenograft models, which opened translational avenues for targeting VEGF-VEGFR signaling [5]. These therapeutic agents can be broadly categorized into those which target the VEGF ligand and those which inhibit the cell surface receptor [6]. VEGFR-TKIs is a class of small molecules targeted therapies that can selectively inhibit the phosphorylation of tyrosine kinase receptors, thereby suppressing tumor angiogenesis. Recently, the National Comprehensive Cancer Network guidelines have listed several VEGFR-TKIs (e.g., sorafenib, lenvatinib) as the first-line treatment for middle-advanced PLC based on their survival benefits observed in different countries and regions [7–12]. Furthermore, alternative VEGFR-TKIs such as regorafenib and apatinib are also being used [13]. Although the advent of targeted therapy for PLC treatment marks a significant epoch, some randomized controlled trials indicated that sorafenib-treated patients had a median overall survival (OS) of only 10.7 months [10], while that of lenvatinib-treated patients was 13.6 months [9]. Moreover, several adverse drug reactions (ADRs) have also been associated with oral targeted drugs, including hypertension, myelosuppression, neurotoxicity, gastrointestinal reactions, and drug resistance, which severely influence the treatment outcome for PLC [14].

In recent years, in addition to targeted therapies, significant breakthroughs in immunotherapy have also been made. A meta-analysis revealed that in patients undergoing immunotherapy (either monotherapy or in combination with other anticancer agents), the pooled odds ratio was 1.67 [95% confidence interval (CI): 1.52–1.84]. Compared to control treatments, immune checkpoint inhibitors have indicated substantially increased rates of achieving complete response [15]. Moreover, gender was observed to influence the efficacy of immune checkpoint inhibitors in cancer patients, with males generally experiencing greater overall benefit after this therapy [16, 17]. It has been indicated that patients with advanced hepatocellular carcinoma (HCC) can benefit from immunotherapy [18]. Furthermore, for treating HCC, many studies have revealed the efficacy of immunotherapy, such as Tremelimumab + Durvalumab and Durvalumab alone [19], as well as the combination of immunotherapy with targeted therapy, such as Atezolizumab + Bevacizumab [20]. The National Comprehensive Cancer Network (2024) guidelines have also approved these treatment strategies.

Despite advancements in treatment options, PLC remains a significant burden for individuals and society. Therefore, alternative therapeutic modalities are urgently required for prolonged survival, enhanced quality of life, and reduced ADRs in PLC cases. New complementary treatment combined with targeted therapy is a possible avenue for exploration. Tradition medicinal preparations (TMPs) comprise traditional prescriptions, extracts, injections, proprietary Chinese medicines, and clinician-prepared decoctions. Several clinical trials have indicated that combining TMPs and loco-regional or systemic therapies increases PLC treatment efficacy [21–23], reduces ADRs [24], and improves patient's quality of life [25], highlighting its potential as a promising approach. Furthermore, mechanistic studies have revealed that nobiletin, a *Citrus aurantium* L. (Rutaceae) extract, inhibits proliferation, promotes apoptosis [26], reverses multidrug resistance [27], and inhibits metastasis [28] in HCC. Moreover, in HCC, ginsenoside Rg3 and Huaier granule have been indicated to inhibit tumor proliferation [29, 30], reduce tumor cell viability [31], decrease tumor metastasis, prolong survival in mouse models [32], and improve the efficacy of the chemotherapeutic drug [33]. Some traditional medicines and their extracts such as the compound Kushen injection [34], artesunate [35], emodin [36], and catalpol [37] can sensitize HCC cells to the anticancer activity of sorafenib, possibly by combating targeted drug resistance and increasing the effect of tyrosine kinase inhibitors (TKIs).

Much literature indicated that VEGFR-TKIs have enhanced the efficacy and reduced adverse drug reactions in PLC patients treated with combined TMPs + VEGFR-TKIs therapy compared to VEGFR-TKIs alone. These findings suggest that compared to current therapeutic modalities, TMPs may serve as effective complementary or alternative treatments for PLC with more favorable risk-benefit profiles. However, because of the limited number of clinical trials investigating the co-treatment of TMPs + VEGFR-TKIs and their small sample sizes, the evidence for its potential use is less convincing. Therefore, our objective is to assess the therapeutic efficacy and safety of TMPs in combination with VEGFR-TKIs for middle-to-advanced PLC, aiming to provide substantial evidence.

## Methodology

### Experimental designs

This study performed a systematic review and meta-analysis based on the guideline of Preferred Reporting Items for Systematic Reviews and Meta-Analysis (PRISMA; S1 File). The protocol has been registered in PROSPERO (CRD42022350634).

## Eligibility criteria

**Investigation profiles.** Randomized controlled trials in English or Chinese languages were selected for this study.

**Patients.** This study included patients diagnosed with middle-advanced PLC according to pathological, cytological, and/or imaging diagnosis criteria. There were no restrictions on patient sex or age.

**Interventions.** There were two cohorts in this study; the experimental cohort received TMPs combined with VEGFR-TKIs, while the control cohort was administered with the same VEGFR-TKIs regimen without TMPs. All formulations or administrations of TMPs, such as decoction, pill, granule, and injection were included. Furthermore, all studies with a minimum treatment course of four weeks were included.

**Primary outcomes.** The primary outcome for treating tumor progression was measured through objective response rate (ORR): complete response (CR) and partial response (PR), together with disease control rate (DCR): CR + PR + stable disease (SD). Outcomes were evaluated before the trial started and after the follow-up ended. For assessment criteria, the guidelines of Response Evaluation Criteria in Solid Tumors (RECIST) [38] and the World Health Organization (WHO) [39] were followed.

**Secondary outcomes.** The secondary outcomes included one-year OS, quality-of-life, level of alpha-fetoprotein (AFP), and ADRs. The Karnofsky Performance Status (KPS) was employed to assess the quality of life. Furthermore, ADR incidences were evaluated for liver dysfunction, proteinuria, hypertension, hand-foot skin reactions, gastrointestinal reactions, myelosuppression, fatigue, and rash. Secondary outcomes were also evaluated before the trial started and after the follow-up ended.

## Exclusion criteria

Studies with (1) incomplete data that could not be used for further analyses, (2) duplicated data, (3) lack of a suitable control cohort, (4) uncertain tumor stage, and (5) not consistent TMPs use in a trial were excluded.

## Literature survey

All randomized controlled trials, published in both Chinese and English, were searched from January 1, 2000 until April 12, 2024. The scientific investigation repositories employed included PubMed, EMBASE, Cochrane Central Register of Controlled Trials (CENTRAL), Turning Research into Practice (TRIP), Latin American and Caribbean Health Sciences Literature (LILACS), Alt HealthWatch, China National Knowledge Infrastructure (CNKI), Chinese Biomedical Literature Database (CBM), Wangfang Datasets Knowledge Service Platform, Chinese Scientific Journal Database (VIP database), clinicaltrials.gov, and Chinese Clinical Trial Registry.

Search terms employed for English databases included: "cancer*", "carcinoma", "neoplasms", "hepatocellular", "traditional medicine", "complementary therapies", "Chinese herbal", "herbal medicine", "Sorafenib", "target therapy", "Vascular Endothelial Growth Factor", "lenvatinib", "apatinib", "regorafenib", together with "random". For Chinese repositories, synonymous Chinese terms were utilized (S2 File indicates full search strategy). References from related studies were also reviewed to retrieve additional studies. This investigation was conducted by two assessors and any dispute was settled by a third reviewer [40].

## Published studies screening and dataset extraction

Study eligibility was independently assessed by two assessors and any disputes were settled by consensus conversation with a third assessor. The following information was extracted: 1. General [title, author, year(s)]; 2. Methodology (study design, baseline comparability, randomization and blinding, loss of follow-up, and selective reporting); 3. Patients (diagnostic criteria, sample size, age, sex, stage of PLC); 4. Intervention (treatment regimens, composition of TMPs, drug dosage, delivery route, therapeutic timeframe); and 5. Investigation endpoints.

## Methodology quality evaluation

The selected studies were independently evaluated by two reviewers and any disagreement was resolved by a third reviewer. The quality of these studies was assessed using the Cochrane risk-of-bias tool covering seven domains: random sequence generation, allocation concealment, blinding of participants and personnel, blinding of outcome assessment, incomplete outcome data, selective reporting, and other biases. Each domain was scored for low, unclear, or high risk of bias [40].

## Datasets analyses

Meta-analysis was performed by two reviewers independently using the RevMan5.4 and Stata17 software. Using the risk ratio (RR), which represents the ratio of the studied outcomes between the co-treatment group and the mono treatment group, along with 95% confidence intervals (CI), dichotomous variables were identified. For continuous variables, the standardized mean difference (SMD) with 95% CI was employed.

The heterogeneity of the selected trials was evaluated and those with $I^2 < 75\%$ (defining no significant statistical heterogeneity) were included for meta-analysis. Furthermore, data from trials with $I^2 \leq 25\%$ were pooled with a fixed-effect model (FEM). Trials with $25\% < I^2 < 75\%$ were assessed for heterogeneity sources *via* the sensitive and sub-cohort analyses. Studies that achieved $I^2 \leq 25\%$ were evaluated through FEM, while those that did not were analyzed through a random-effects model (REM) within meta-analyses. Moreover, data from significantly heterogeneous trials ($I^2 \geq 75\%$) were not pooled because such large heterogeneity cannot be explained *via* sub-cohort analyses. To eliminate the possible publication bias, funnel-plot, Begg's test, and Egger's test were performed for meta-analysis with $\geq 10$ trials.

Subgroup analyses were conducted in accordance with the target drug regimen, course of treatment, KPS score, and form of TMPs (e.g., decoction, granule, pill, and powder). Sensitivity assessment indicated whether primary analyses of trials with and without high risks are sufficiently robust, and if the FEM or REM-based meta-analyses are equally reliable.

## Proof quality evaluation

Proof quality from study endpoints was independently evaluated by two reviewers based on the GRADE (Grading of Recommendations Assessment Development and Evaluation) criteria [41], which included 5 domains: risk of bias, inconsistency of trials, indirectness of evidence, imprecision of results, and publication bias. For each domain, four levels of scoring (high, moderate, low, and very low) were applied. Any dispute was settled through a third evaluator.

## Sensitivity analyses of the administered TMPs interventions

To explore the effect of mono or co-treatment of traditional medicines on the efficacy of VEGFR-TKIs in PLC patients, sensitivity analyses of multi-component TMPs were performed based on the composition of TMPs. The rationale of this approach was provided by Chen *et al*.

that the anti-tumor properties of a particular traditional medicine can be reflected in pooled results of the studies on this medicine [42]. In this research, some multi-component TMPs and their several formulations or combinations were studied. Thus, the increased efficacy of medicine by the addition of TMPs with common traditional medicines can be explored by the pooled RRs in study cohorts. Moreover, it may identify specific traditional medicine combinations that contribute the most to the treatment of PLC.

This approach involved producing a matrix of pooled RRs from multiple sub-cohort studies *via* a multilevel procedure. Level 1) all studies using a single identical traditional medicine were regarded as a sub-cohort and the pooled RR and $I^2$ for ORR and DCR were determined. Dataset outcomes were compiled in descending order and any considerable dataset outcomes were recorded. Traditional medicines from sub-cohorts with no major influence on ORR/DCR and major heterogeneity ($I^2 > 30\%$) were excluded from the study. Level 2, the same pair of traditional medicines within TMP interventions were identified as sub-cohorts, and the RRs were calculated. Level 3, combinations of three medicinal plants were identified until no eligible combinations were available. Traditional medicines that met the following inclusion criteria [42] were selected for further analysis:

1. Higher sub-cohort RRs for experimental *vs*. control.

2. Higher sub-cohort RRs than those within the total pool.

3. No significant heterogeneity ($I^2 \leq 30\%$).

4. The sub-cohort RRs were significant at multilevel combinations.

## Results

### Literature search

A total of 2577 potential studies were identified (Fig 1). After removing the duplicated articles, 1984 investigations remained (S3 File). Title screening excluded 1678 articles and 306 remained after abstract screening. Finally, after the full-text screening, 53 articles remained, of which 26 met the inclusion criteria, and therefore were selected for further analyses. Dataset collection and quantitative synthesis were performed for the 26 eligible studies.

### Profiling of analyzed studies

This investigation included 26 randomized controlled trials [43–68] comprising 1678 patients (841 experimental and 837 control). Table 1 reflects the basic profiles of the selected investigations. The sample size of the studies was between 38–119. In the TMP cohort, 15 studies [43, 44, 46–50, 52, 54, 55, 57, 59, 61, 63, 66] used decoction, 6 utilized granules [53, 58, 62, 64, 65, 68], 2 used pill [45, 51], 1 employed powder [67], 1 used bolus [60], and only 1 study utilized injection [56]. All studies used the oral route of delivery, except for one study [56] which used intravenous administration. The general principles of TMPs included heat dissipation toxin removal, tonifying, and blood activation, liver soothing, depression relief, qi tonifying, and spleen fortifying. In total, 24 TMPs were included, involving 105 types of traditional medicines. Table 2 presents the specific formulations, medicines, dosages, and drug quality control. The frequency of use of the top 20 traditional medicines is provided in Fig 2. For VEGFR-TKI regimens, 15 studies [43–45, 47, 51, 52, 55, 56, 58, 60, 62, 63, 65–67] used sorafenib, 6 used apatinib [46, 49, 54, 57, 61, 64], 4 used lenvatinib [48, 50, 59, 68], and 1 used regorafenib [53]. All VEGFR-TKIs were administered orally. Moreover, 21 studies [43–46, 50, 52–65, 67, 68]

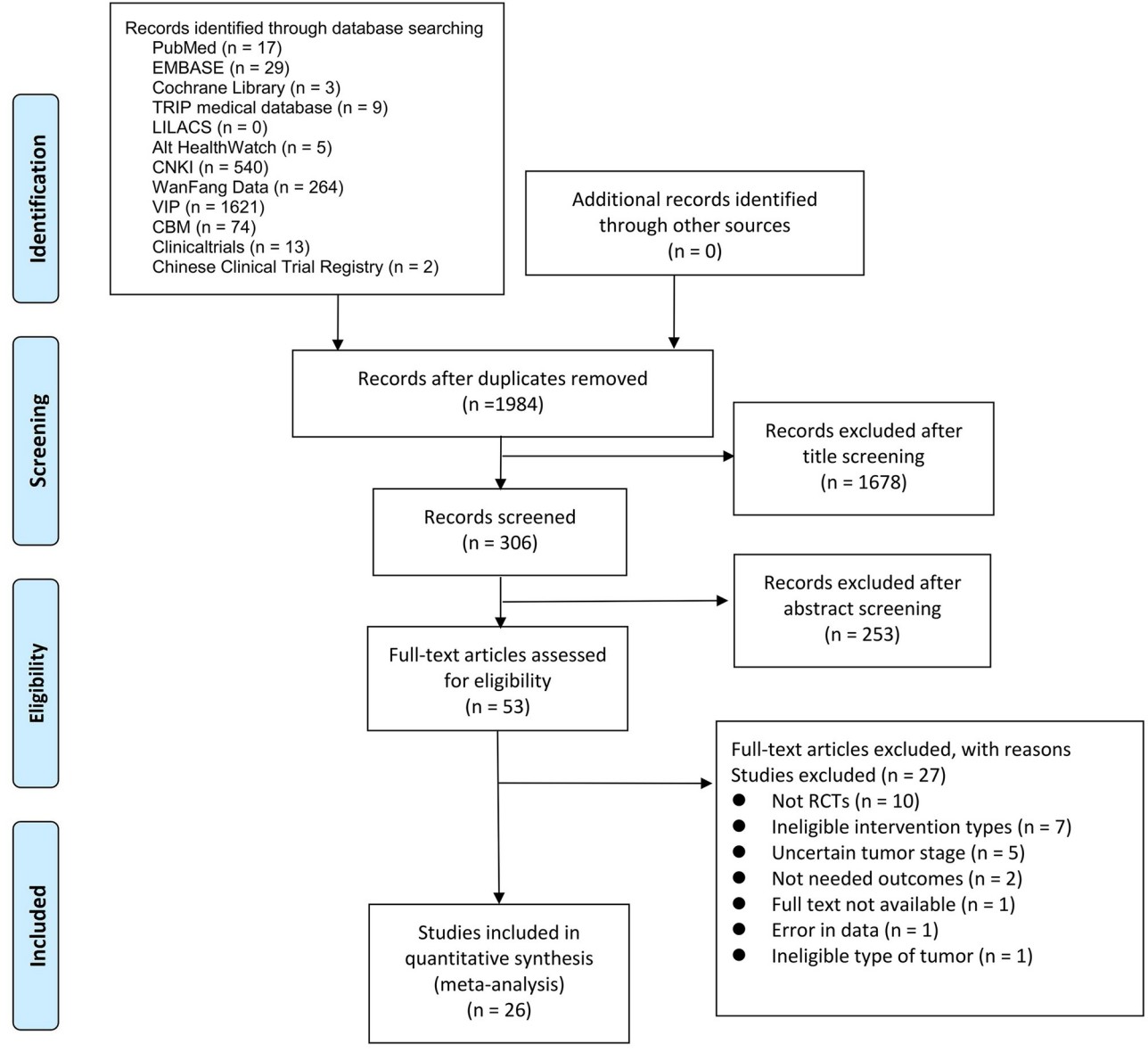

**Fig 1. Flow chart illustrating the literature survey strategy.**

reported ORR and DCR based on the RECIST or WHO guidelines. Eight studies [43, 44, 47, 58, 60, 65–67] reported one-year OS, ten [43, 47, 48, 52, 58, 59, 61, 62, 64, 68] reported quality of life according to the Karnofsky Performance Status (KPS), fourteen [44, 46, 48, 50, 52–54, 58, 60–62, 64, 67, 68] indicated the AFP levels, and 20 studies [44–46, 48–54, 56, 58, 60–62, 64–68] revealed ADRs.

## Investigation quality evaluation

To determine dataset quality within 26 enrolled studies, their risk of bias was assessed (Fig 3, S4 File). The "random sequence generation" domain was evaluated using the random number table method and revealed 13 studies [44, 46–48, 50, 53–57, 61, 62, 64] with low risk of bias

**Table 1. Characteristics of studies on middle-advanced PLC with TMPs and VEGFR-TKIs.**

| Study ID | Simple Size (T/C) | M/F | QoL score | Stage | Age(T/C), median or mean±SD | Interventions | | | Course of Treatment | Outcomes |
|---|---|---|---|---|---|---|---|---|---|---|
| | | | | | | TMPs | Drug delivery | Targeted drugs Regimen | | |
| DuanKN 2018 | 23/22 | T:17/6 C:15/7 | / | III, IV | T(65.66 ±2.12) C(65.70 ±2.23) | Self prescribed Decoction, qd | Orally | Sorafenib,400mg, bid | 2m | 123 |
| FangHS 2015 | 30/30 | T:17/13 C:16/14 | KPS>60 | III:7 IVa:6 IVb:47 | T(59.50 ±11.23) C(57.90 ±10.59) | Shentao Ruangan Decoction,150-200mL, qd | Orally | Sorafenib,400mg, bid | 1m | 1245 |
| FengLH 2012 | 30/29 | T:25/5 C:25/4 | KPS>60 | II, III, IV | T:56.4 C:55.8 | Hua Chan Su Pills:1.2g, tid | Orally | Sorafenib,400mg, bid | 2m | 15 |
| GaoYY 2022 | 47/47 | T:29/18 C:31/16 | / | IV:94 | T(62.17 ±6.03) C(61.51 ±6.28) | Peiyuan Kangai Decoction, 200mL, bid | Orally | Apatinib,500mg, qd | 2m | 145 |
| JinZ 2022 | 29/29 | T:18/12 C:20/10 | KPS≥30 | IIb, IIIa, IIIb | T(56.7±4.6) C(56.4±4.3) | Huisheng oral liquid, 10mL, tid | Orally | Lenvatinib, >60kg~12mg, qd <60kg~8mg, qd | 2m | 145 |
| HanGM 2021 | 29/29 | T:10/19 C:13/16 | / | III, IV | T(53.36 ±10.25) C(53.73 ±10.41) | Jiawei Chaihu Biejia Decoction, bid | Orally | Sorafenib,400mg, bid | 2m | 23 |
| JiangXQ 2022 | 37/38 | T:20/17 C:22/16 | KPS≥30 | III:21 IV:54 | T(61.21 ±4.28) C(61.28 ±4.30) | Chaihu Shugan Huayu Decoction, 200mL, bid | Orally | Lenvatinib, >60kg~12mg, qd <60kg~8mg, qd d1-d21,21d/C | 3cycles | 345 |
| JinJ 2020 | 29/28 | T:23/6 C:23/5 | KPS≥60 | IIIa:15 IIIb:18 IV:24 | unclear | Ganji Decoction, bid | Orally | Apatinib, 250mg, qd, d1-d28,28d/C | 2C | 5 |
| LiDQ 2018 | 30/30 | 36/24 | ECOG 0–2 | III:20 IV:40 | (51.12 ±10.01) | Wu Zhi Pills: 0.93g, tid | Orally | Sorafenib,400mg, bid | 1m | 5 |
| LiuJP 2018 | 33/33 | T:16/17 C:18/15 | KPS>60 | III, IV | T(56.6±5.5) C(56.9±5.5) | Qinghuo Tongluo Decoction, 200mL, tid | Orally | Sorafenib,400mg, bid | 3m | 1345 |
| MaJR 2022 | 55/55 | T:30/25 C:32/23 | / | IIb:30 IIIa:67 IIIb:13 | T(54.96 ±9.15) C(55.31 ±9.20) | Huazhi Rougan Granule, 8g, tid | Orally | Regorafenib,160mg, qd, d1-d21,28d/C | 3cycles | 145 |
| MaYK 2018 | 20/20 | T:17/3 C:15/5 | KPS≥70 | BCLC: C:33 D:7 | T(58.8±7.7) C(59.6±9.3) | Jianpi Jiedu Decoction, 150mL, bid | Orally | Apatinib,250mg, qd | 2m | 145 |
| QiaoCX 2015 | 20/20 | T:14/6 C:19/1 | / | II, III, IV | T:median:49 C: median:46.5 | Fuzheng Guben Decoction, 100mL, bid | Orally | Sorafenib,400mg, bid | 2m | 1 |
| SunJ 2022 | 60/59 | T:32/28 C:34/25 | KPS T(79.63 ±10.12) C(80.48 ±10.23) | III:62 IV:57 | T(53.96 ±4.58) C(54.15 ±4.63) | Elemene Injection, 0.6g, qd | Intravenously | Sorafenib,400mg, bid | 6m | 15 |

*(Continued)*

**Table 1.** (Continued)

| Study ID | Simple Size (T/C) | M/F | QoL score | Stage | Age(T/C), median or mean±SD | Interventions | | | Course of Treatment | Outcomes |
|---|---|---|---|---|---|---|---|---|---|---|
| | | | | | | TMPs | Drug delivery | Targeted drugs Regimen | | |
| SunY 2019 | 30/30 | T:17/13 C:16/14 | KPS>60 | II:15 III:21 IV:24 | T(65.69 ±1.38) C(65.53 ±1.25) | Jiawei Yiguan Decoction, 120mL, bid | Orally | Apatinib,500mg, qd | >1m | 1 |
| TangYF 2018 | 57/56 | T:40/17 C:41/15 | / | III, IV | T(51.97 ±7.41) C(52.34 ±7.27) | Huai Er Granule: 20g, tid | Orally | Sorafenib,400mg, bid | 2m | 12345 |
| TuXL 2021 | 30/30 | T:19/11 C:18/12 | / | BCLC: C:60 | T(53.93 ±5.29) C(53.56 ±6.12) | Jianpi Yangang Jiedu Decoction, 150mL, bid | Orally | Lenvatinib, >60kg~12mg, qd <60kg~8mg, qd | 2m | 13 |
| WangGT 2016 | 18/20 | 24/14 | / | IVa:36 IVb:2 | (45.2±4.6) | Songxiang Bolus,20g, bid | Orally | Sorafenib,400mg, bid | Average duration of medication 157.4d | 1245 |
| WuYW 2018 | 30/30 | T:27/3 C:28/2 | KPS≥60 | III:16 IV:44 | T(58.04 ±11.41) C(60.8 ±14.11) | Jianpi Rougan Decoction,125mL, bid | Orally | Apatinib,250mg, qd, d1-d28,28d/C | >1cycles | 1345 |
| YangCJ 2021 | 27/26 | T:19/8 C:20/6 | KPS≥60 | BCLC: B:18 C:35 | T(56.96 ±9.89) C(54.88 ±10.25) | Jiawei Xiao Chaihu Granule, tid | Orally | Sorafenib,400mg, bid | 2m | 1345 |
| YuJF 2021 | 33/32 | T:27/6 C:27/5 | ECOG 0–2 | BCLC: C:65 | T(47.33 ±12.35) C(48.82 ±12.34) | Fuling Sini Decoction | Orally | Sorafenib,400mg, bid | >1m | 1 |
| Zhang QH 2019 | 30/30 | T:23/7 C:21/9 | KPS>60 | III, IV | T(49.63 ±9.06) C(52.57 ±8.92) | Huai Er Granule: 20g, tid | Orally | Sorafenib,400mg, bid | 2m | 125 |
| ZhangZ 2019 | 28/28 | T:18/10 C:20/8 | / | BCLC: B, C | T(55.93 ±7.03) C(56.68 ±6.56) | Yiqi Huayu Jiedu Decoction, bid | Orally | Sorafenib,400mg, bid | Until the patient had grade 3 or 4 adverse reactions or died | 25 |
| ZhanLH 2022 | 30/30 | T:18/12 C:21/9 | KPS≥60 | T: IIb:15, IIIa:15 C: IIb:13, IIIa:17 | T(58.83 ±10.93) C(58.80 ±12.93) | Chai Shao Granule, bid | Orally | Apatinib,750mg, qd | 3m | 1345 |
| ZhouFJ 2020 | 30/30 | T:17/13 C:18/12 | KPS≥60 | III, IV | T(59.51 ±11.22) C(59.80 ±10.58) | Jiajian Sanjia Powder, 10g, tid | Orally | Sorafenib,400mg, bid, d1-d28,28d/C | >1cycles | 1245 |
| ZhuX 2023 | 26/26 | T:21/5 C:22/4 | KPS≥60 | III:40 IV:12 | T(60.00 ±9.45) C(61.38 ±9.19) | Danzhi Xiaoyao Granule, 300ml, bid | Orally | Lenvatinib, >60kg~12mg, qd <60kg~8mg, qd | 2m | 135 |

while the remaining studies indicated unclear risk due to the lack of specific methods. For "allocation concealment" assessed with the envelope method, only one study [44] had a low risk of bias, while the remaining indicated unclear risk because they did not report the blinding methods. Furthermore, all studies had a low risk of bias for "incomplete outcome data" since

**Table 2. The formulation, medicines and dosage, and control of drug quality.**

| Study | Formulation | Source | Species, concentration | Quality control reported? (Y/N) | Chemical analysis reported? (Y/N) |
|---|---|---|---|---|---|
| DuanKN 2018 | Self prescribed Decoction | Shanxi Taiyuan Hospital of Traditional Chinese Medicine | *Dioscorea oppositifolia* L. [Dioscoreaceae] 30g, *Akebia trifoliata* (Thunb.) Koidz. [Lardizabalaceae] 30g, *Astragalus mongholicus* Bunge [Fabaceae] 20-30g, Galli Gigerii Endothelium Corneum 20g, *Rehmannia glutinosa* (Gaertn.) DC. [Orobanchaceae] 15-20g, *Areca catechu* L. [Arecaceae] 15g, *Solanum nigrum* L. [Solanaceae] 15g, *Scutellaria barbata* D.Don [Lamiaceae] 15g, *Paeonia lactiflora* Pall. [Paeoniaceae] 15g, *Atractylodes macrocephala* Koidz. [Asteraceae] 10g, *Eupolyphaga sinensis* Walker 5g, *Scolopendra subspinipes mutilans* 2. | Y—Quality controlled by Shanxi Taiyuan Hospital of Traditional Chinese Medicine | N |
| FangHS 2015 | Shentao Ruangan Decoction | The first affiliated hospital of Guangzhou university of Chinese medicine | *Panax ginseng* C.A. Mey.15g, *Prunus persica* (L.) Batsch [Rosaceae]10g, *Angelica sinensis* (Oliv.) Diels [Apiaceae]10g, *Agrimonia pilosa* var. nepalensis (D.Don) Nakai [Rosaceae]30g, *Cremastra appendiculata* (D.Don) Makino [Orchidaceae]15g, *Scutellaria barbata* D.Don [Lamiaceae]15g, *Atractylodes macrocephala* Koidz. [Asteraceae]15g, *Glycyrrhiza glabra* L. [Fabaceae] 6g. | Y—Quality controlled by the first affiliated hospital of Guangzhou university of Chinese medicine | N |
| FengLH 2012 | Hua Chan Su Pills | China resources Jinchan Pharmaceuticals Co. Ltd | Bufonis Corium. | Y- Prepared according to The Pharmacopoeia of the People's Republic of China | Y- TLC |
| GaoYY 2022 | Peiyuan Kangai Decoction | Characteristic Medical Center of the Chinese Armed Police Force | *Astragalus mongholicus* Bunge [Fabaceae] 30g, *Atractylodes macrocephala* Koidz. [Asteraceae]15g, *Ligustrum lucidum* W.T.Aiton [Oleaceae]15g, *Scrophularia ningpoensis* Hemsl. [Scrophulariaceae] 15g, *Eucommia ulmoides* Oliv. [Eucommiaceae]15g, *Hordeum vulgare* L. [Poaceae]15g, *Cullen corylifolium* (L.) Medik. [Fabaceae]15g, *Pinellia ternata* (Thunb.) Makino [Araceae]15g, *Citrus × aurantium* L.10g, *Wurfbainia villosa* (Lour.) Skornick. & A.D. Poulsen [Zingiberaceae]10g, *Curcuma aromatica* Salisb. [Zingiberaceae]10g, *Eupolyphaga sinensis* Walker 10g, *Gypsophila vaccaria* (L.) Sm. [Caryophyllaceae] 12g. | Y—Quality controlled by Characteristic Medical Center of the Chinese Armed Police Force | N |
| JinZ 2022 | Huisheng oral liquid | Chengdu Di'ao Group Tianfu Pharmaceutical Stock Co.,Ltd. | *Panax ginseng* C.A.Mey., *Angelica sinensis* (Oliv.) Diels [Apiaceae], *Paeonia lactiflora* Pall. [Paeoniaceae], *Curcuma longa* L. [Zingiberaceae], *Tetradium ruticarpum* (A.Juss.) T.G.Hartley [Rutaceae], *Alisma plantago-aquatica* subsp. orientale (Sam.) Sam. [Alismataceae], *Rheum palmatum* L. [Polygonaceae], *Cyperus rotundus* L. [Cyperaceae], *Conioselinum anthriscoides* 'Chuanxiong' [Apiaceae], *Sparganium stoloniferum* (Buch.-Ham. ex Graebn.) Buch.-Ham. ex Juz. [Typhaceae], *Prunus persica* (L.) Batsch [Rosaceae], *Boswellia sacra* Flück. [Burseraceae], *Commiphora myrrha* (T.Nees) Engl. [Burseraceae], *Trionyx sinensis* Wiegmann shell, *Syzygium aromaticum* (L.) Merr. & L.M.Perry [Myrtaceae], *Corydalis yanhusuo* (Y.H.Chou & Chun C.Hsu) W.T.Wang ex Z.Y.Su & C.Y.Wu [Papaveraceae], *Rehmannia glutinosa* (Gaertn.) DC. [Orobanchaceae]. | Y- Prepared according to The Pharmacopoeia of the People's Republic of China | Y- TLC, HPLC |

*(Continued)*

**Table 2.** (Continued)

| Study | Formulation | Source | Species, concentration | Quality control reported? (Y/N) | Chemical analysis reported? (Y/N) |
|---|---|---|---|---|---|
| HanGM 2021 | Jiawei Chaihu Biejia Decoction | Yuebei people's Hospital | *Pseudostellaria heterophylla* (Miq.) Pax [Caryophyllaceae] 12g, *Panax japonicus* (T.Nees) C.A.Mey. [Araliaceae]12g, *Atractylodes macrocephala* Koidz. [Asteraceae]12g, *Smilax glabra* Roxb. [Smilacaceae]15g, *Crataegus monogyna* Jacq. [Rosaceae] 9g, Galli Gigerii Endothelium Corneum 15g, *Pinellia ternata* (Thunb.) Makino [Araceae] 9g, *Citrus × aurantium* L. 5g, *Bergenia purpurascens* (Hook.f. & Thomson) Engl. [Saxifragaceae]15g, *Strobilanthes cusia* (Nees) Kuntze [Acanthaceae] 15g, *Smilax glabra* Roxb. [Smilacaceae] 15g, *Hypericum japonicum* Thunb. [Hypericaceae] 15g, *Bursa bursa-pastoris rhomboidea* Shull. 15g, *Prunella vulgaris* L. [Lamiaceae] 9g, Crassostrea 15g, *Arisaema amurense* Maxim. [Araceae] 9g, *Curcuma aromatica* Salisb. [Zingiberaceae]9g, Gekko gecko 5g, *Ageratum conyzoides* L. [Asteraceae] 6g, *Astragalus mongholicus* Bunge [Fabaceae] 15g, *Angelica sinensis* (Oliv.) Diels [Apiaceae] 9g. | Y—Quality controlled by Yuebei people's Hospital | N |
| JiangXQ 2022 | Chaihu Shugan Huayu Decoction | Baoji People's Hospital | *Salvia miltiorrhiza* Bunge [Lamiaceae] 20g, *Paeonia lactiflora* Pall. [Paeoniaceae] 20g, Trionyx sinensis Wiegmann shell 20g, Crassostrea 20g, *Bupleurum chinese* DC. [Apiaceae] 15g, *Scrophularia ningpoensis* Hemsl. [Scrophulariaceae] 15g, *Fritillaria thunbergii* Miq. [Liliaceae] 15g, *Conioselinum anthriscoides* 'Chuanxiong' [Apiaceae] 15g, *Citrus × aurantium* L. [Rutaceae] 15 g, *Citrus × aurantium* L. [Rutaceae] 10g, *Cyperus rotundus* L. [Cyperaceae] 10g, *Santalum album* L. [Santalaceae] 6g, *Panax notoginseng* (Burkill) F.H.Chen [Araliaceae] 5g, *Glycyrrhiza glabra* L. [Fabaceae] 8g. | Y—Quality controlled by Baoji People's Hospital | N |
| JinJ 2020 | Ganji Decoction | Guangdong Integrated Traditional Chinese and Western Medicine Hospital | *Codonopsis pilosula* (Franch.) Nannf. [Campanulaceae]15g, *Astragalus mongholicus* Bunge [Fabaceae]15g, *Bupleurum chinese* DC. [Apiaceae]10g, *Paeonia lactiflora* Pall. [Paeoniaceae]15g, *Curcuma aromatica* Salisb. [Zingiberaceae]15g, *Atractylodes macrocephala* Koidz. [Asteraceae]15g, *Scrophularia ningpoensis* Hemsl. [Scrophulariaceae]10g, *Lycium barbarum* L. [Solanaceae]15g, Galli Gigerii Endothelium Corneum 15g, *Akebia trifoliata* (Thunb.) Koidz. [Lardizabalaceae]10g, Trionyx sinensis Wiegmann shell 15g, *Curcuma longa* L. [Zingiberaceae]15g. | Y—Quality controlled by Guangdong Integrated Traditional Chinese and Western Medicine Hospital | N |
| LiDQ 2018 | Wu Zhi Pills | Guangxi Fanglue Pharmaceutical Group Co. Ltd. | Schisantherin A | Y- Prepared according to The Pharmacopoeia of the People's Republic of China | Y- TLC |
| LiuJP 2018 | Qinghuo Tongluo Decoction | Wuhan Central Hospital | *Scleromitrion diffusum* (Willd.) R.J.Wang [Rubiaceae], *Scutellaria barbata* D.Don [Lamiaceae], *Paris polyphylla* Sm. [Melanthiaceae], *Reynoutria japonica* Houtt. [Polygonaceae], *Swertia chirayita* (Roxb.) H.Karst. [Gentianaceae], *Manis pentadactyla* Linnaeus, *Scolopendra subspinipes mutilans*, Scorpion, *Eupolyphaga sinensis* Walker, *Tripidium arundinaceum* (Retz.) Welker, Voronts. & E.A.Kellogg [Poaceae]. | Y—Quality controlled by Wuhan Central Hospital | N |

(*Continued*)

**Table 2.** (Continued)

| Study | Formulation | Source | Species, concentration | Quality control reported? (Y/N) | Chemical analysis reported? (Y/N) |
|---|---|---|---|---|---|
| MaJR 2022 | Huazhi Rougan Granule | Shandong New Time Pharmaceutical Co. Ltd. | *Swertia chirayita* (Roxb.) H.Karst. [Gentianaceae], *Senna tora* (L.) Roxb. [Fabaceae], *Rheum palmatum* L. [Polygonaceae], *Crotalaria albida* B. Heyne ex Roth [Fabaceae], *Crataegus monogyna* Jacq. [Rosaceae], *Atractylodes macrocephala* Koidz. [Asteraceae], *Citrus × aurantium* L. [Rutaceae], *Glycyrrhiza glabra* L. [Fabaceae], *Lycium barbarum* L. [Solanaceae], *Astragalus mongholicus* Bunge [Fabaceae], *Glycyrrhiza glabra* L. [Fabaceae], *Paeonia lactiflora* Pall. [Paeoniaceae]. | Y- Prepared according to The Pharmacopoeia of the People's Republic of China | Y- TLC, HPLC |
| MaYK 2018 | Jianpi Jiedu Decoction | Shanghai Xuhui District Central Hospital | *Pseudostellaria heterophylla (Miq.) Pax [Caryophyllaceae] 12g, Panax japonicus (T.Nees) C. A.Mey. [Araliaceae]12g, Atractylodes macrocephala Koidz. [Asteraceae]12g, Smilax glabra Roxb. [Smilacaceae]15g, Crataegus monogyna Jacq. [Rosaceae] 9g, Galli Gigerii Endothelium Corneum 15g, Pinellia ternata (Thunb.) Makino [Araceae] 9g, Citrus × aurantium L. 5g, Bergenia purpurascens (Hook.f. & Thomson) Engl. [Saxifragaceae]15g, Strobilanthes cusia (Nees) Kuntze [Acanthaceae] 15g, Smilax glabra Roxb. [Smilacaceae] 15g, Hypericum japonicum Thunb. [Hypericaceae] 15g, Bursa bursa-pastoris (L.) Medik. [Brassicaceae] 15g, Prunella vulgaris L. [Lamiaceae]9g, Crassostrea 15g, Arisaema amurense Maxim. [Araceae] 9g, Curcuma aromatica Salisb. [Zingiberaceae]9g, Gekko gecko 5g, Ageratum conyzoides L. [Asteraceae] 6g, Astragalus mongholicus Bunge [Fabaceae] 15g, Angelica sinensis (Oliv.) Diels [Apiaceae] 9g.* | Y—Quality controlled by Shanghai Xuhui District Central Hospital | N |
| QiaoCX 2015 | Fuzheng Guben Decoction | He'nan Institute of Traditional Chinese Medicine Research Hospital | *Panax ginseng C.A.Mey., Ganoderma lucidum, Bufonis Corium* | Y—Quality controlled by He'nan Institute of Traditional Chinese Medicine Research Hospital | N |
| SunJ 2022 | Elemene Injectable Emulsion | Dalian Huali Jingang Pharmaceutical Co. Ltd. | *β-elemene* | Y- Prepared according to The Pharmacopoeia of the People's Republic of China | Y- GC, HPLC |
| SunY 2019 | Jiawei Yiguan Decoction | Huizhou Hospital of Traditional Chinese Medicine | *Rehmannia glutinosa (Gaertn.) DC. [Orobanchaceae]15g, Lycium barbarum L. [Solanaceae]15g, Glehnia littoralis (A.Gray) F. Schmidt ex Miq. [Apiaceae]15g, Ophiopogon japonicus (Thunb.) Ker Gawl. [Asparagaceae]15g, Angelica sinensis (Oliv.) Diels [Apiaceae]10g, Melia azedarach L. [Meliaceae]10g, Eupolyphaga sinensis Walker 5g, Prunus persica (L.) Batsch [Rosaceae] 10g, Solanum nigrum L. [Solanaceae]5g.* | Y—Quality controlled by Huizhou Hospital of Traditional Chinese Medicine | N |
| TangYF 2018 | Huai Er Granule | Qidong Gaitianli Medicines Co. Ltd. | *Pseudostellaria heterophylla (Miq.) Pax [Caryophyllaceae]* | Y- Prepared according to The Pharmacopoeia of the People's Republic of China | Y- TLC |

(*Continued*)

**Table 2.** (Continued)

| Study | Formulation | Source | Species, concentration | Quality control reported? (Y/N) | Chemical analysis reported? (Y/N) |
|---|---|---|---|---|---|
| TuXL 2021 | Jianpi Yanggan Jiedu Decoction | Ningbo Hospital of Traditional Chinese Medicine | *Astragalus mongholicus Bunge [Fabaceae] 30g, Codonopsis pilosula (Franch.) Nannf. [Campanulaceae] 20g, Atractylodes macrocephala Koidz. [Asteraceae] 20g, Coix lacryma-jobi L. [Poaceae] 30g, Smilax glabra Roxb. [Smilacaceae] 15g, Lycium barbarum L. [Solanaceae] 15g, Ligustrum lucidum W.T.Aiton [Oleaceae] 15g, Scutellaria baicalensis Georgi [Lamiaceae] 12g, Forsythia suspensa (Thunb.) Vahl [Oleaceae] 15, Lonicera japonica Thunb. [Caprifoliaceae] 15g, Fritillaria thunbergii Miq. [Liliaceae] 12g, Trionyx sinensis Wiegmann shell 20g, Isodon rubescens (Hemsl.) H.Hara [Lamiaceae] 30g.* | Y—Quality controlled by Ningbo Hospital of Traditional Chinese Medicine | N |
| WangGT 2016 | Songxiang Bolus | The affiliated hospital of Shanxi university of Chinese medicine | *Abrus melanospermus subsp. melanospermus [Fabaceae]* | Y—Quality controlled by the affiliated hospital of Shanxi university of Chinese medicine | N |
| WuYW 2018 | Jianpi Rougan Decoction | Guangdong Integrated Traditional Chinese and Western Medicine Hospital | *Codonopsis pilosula (Franch.) Nannf. [Campanulaceae] 15g, Galli Gigerii Endothelium Corneum 15g, Lycium barbarum L. [Solanaceae] 15g, Astragalus mongholicus Bunge [Fabaceae] 15g, Akebia trifoliata (Thunb.) Koidz. [Lardizabalaceae] 10g, Paeonia lactiflora Pall. [Paeoniaceae] 15g, Scrophularia ningpoensis Hemsl. [Scrophulariaceae] 10g, Curcuma aromatica Salisb. [Zingiberaceae] 15g, Atractylodes macrocephala Koidz. [Asteraceae] 15g, Bupleurum chinense DC. [Apiaceae] 10g, Curcuma longa L. [Zingiberaceae] 15g, Trionyx sinensis Wiegmann shell 15g, Pinellia ternata (Thunb.) Makino [Araceae] 10g.* | Y—Quality controlled by Guangdong Integrated Traditional Chinese and Western Medicine Hospital | N |
| YangCJ 2021 | Jiawei Xiao Chaihu Granule | Sichuan Neo-Green Pharmaceutical Technology Development Co. Ltd. | *Bupleurum Chinense DC. [Apiaceae] 15g, Scutellaria baicalensis Georgi [Lamiaceae] 10g, Pinellia ternata (Thunb.) Makino [Araceae] 15g, Panax ginseng C.A.Mey. 20g, Paeonia lactiflora Pall. [Paeoniaceae] 15g, Citrus × aurantium L. [Rutaceae] 15g, Citrus × aurantium L. [Rutaceae] 15g, Citrus × aurantium L. 15g, Smilax glabra Roxb. [Smilacaceae] 15g, Atractylodes macrocephala Koidz. [Asteraceae] 15g, Zingiber officinale Roscoe [Zingiberaceae] 10g, Glycyrrhiza glabra L. [Fabaceae] 5g.* | Y- Prepared according to The Pharmacopoeia of the People's Republic of China | Y- HPLC |
| YuJF 2021 | Fuling Sini Decoction | Liuzhou Hospital of Traditional Chinese Medicine | *Smilax glabra Roxb. [Smilacaceae] 40g, Glycyrrhiza glabra L. [Fabaceae] 10g, Zingiber officinale Roscoe [Zingiberaceae] 10g, Raphanus raphanistrum subsp. sativus (L.) Domin [Brassicaceae] 15g, Panax ginseng C.A.Mey. 10g.* | Y—Quality controlled by Liuzhou Hospital of Traditional Chinese Medicine | N |
| ZhangQH 2019 | Huai Er Granule | Qidong Gaitianli Medicines Co. Ltd. | *Pseudostellaria heterophylla (Miq.) Pax [Caryophyllaceae]* | Y- Prepared according to The Pharmacopoeia of the People's Republic of China | Y- TLC |
| ZhangZ 2019 | Yiqi Huayu Jiedu Decoction | Hunan Academy of Traditional Chinese Medicine Affiliated Hospital | *Panax ginseng C.A.Mey. 15g, Atractylodes macrocephala Koidz. [Asteraceae] 15g, Smilax glabra Roxb. [Smilacaceae] 15g, Curcuma aromatica Salisb. [Zingiberaceae]15g, Scutellaria barbata D.Don [Lamiaceae]15g, Astragalus mongholicus Bunge [Fabaceae] 30g, Eupolyphaga sinensis Walker, Paris polyphylla var. yunnanensis (Franch.) Hand. -Mazz. [Melanthiaceae]10g, Gekko gecko 10g, Trionyx sinensis Wiegmann shell 6g, Scorpion 5g.* | Y—Quality controlled by Hunan Academy of Traditional Chinese Medicine Affiliated Hospital | N |

*(Continued)*

**Table 2.** (Continued)

| Study | Formulation | Source | Species, concentration | Quality control reported? (Y/N) | Chemical analysis reported? (Y/N) |
|---|---|---|---|---|---|
| ZhanLH 2022 | Chai Shao Granule | The first affiliated hospital of Guangxi University of Chinese Medicine | *Bupleurum Chinense DC. [Apiaceae], Pinellia ternata (Thunb.) Makino [Araceae], Scutellaria baicalensis Georgi [Lamiaceae], Zingiber officinale Roscoe [Zingiberaceae], Glycyrrhiza glabra L. [Fabaceae], Ziziphus jujuba Mill. [Rhamnaceae], Codonopsis pilosula (Franch.) Nannf. [Campanulaceae], Paeonia lactiflora Pall. [Paeoniaceae], Angelica sinensis (Oliv.) Diels [Apiaceae], Alisma plantago-aquatica subsp. Orientale (Sam.) Sam. [Alismataceae], Smilax glabra Roxb. [Smilacaceae], Conioselinum anthriscoides 'Chuanxiong' [Apiaceae], Atractylodes macrocephala Koidz. [Asteraceae].* | Y—Quality controlled by the first affiliated hospital of Guangxi University of Chinese Medicine | N |
| ZhouFJ 2020 | Jiajian Sanjia Powder | Handan Hospital of Traditional Chinese Medicine | *Trionyx sinensis Wiegmann shell 10g, Astragalus mongholicus Bunge [Fabaceae] 10g, Angelica sinensis (Oliv.) Diels [Apiaceae] 10g, Manis pentadactyla Linnaeus 10g, Eupolyphaga sinensis Walker 10g, Beauveria bassiana (Bals.) Vuillant 10g, Prunus persica (L.) Batsch [Rosaceae] 10g, Bupleurum chinense DC. [Apiaceae] 10g.* | Y—Quality controlled by Handan Hospital of Traditional Chinese Medicine | N |
| ZhuX 2023 | Danzhi Xiaoyao Granule | Cangzhou Hospital of Integrated Chinese and Western Medicine | *Paeonia × suffruticosa Andrews [Paeoniaceae] 9g, Gardenia jasminoides J.Ellis [Rubiaceae] 9g, Angelica sinensis (Oliv.) Diels [Apiaceae] 12g, Paeonia lactiflora Pall. [Paeoniaceae] 12g, Bupleurum Chinense DC. [Apiaceae] 12g, Smilax glabra Roxb. [Smilacaceae] 15g, Atractylodes macrocephala Koidz. [Asteraceae] 15g, Glycyrrhiza glabra L. [Fabaceae] 6g, Zingiber officinale Roscoe [Zingiberaceae] 6g, Mentha spicata L. [Lamiaceae] 6g, Trionyx sinensis Wiegmann shell 12g, Fritillaria thunbergii Miq. [Liliaceae] 12g, Curcuma aromatica Salisb. [Zingiberaceae] 9g.* | Y—Quality controlled by Cangzhou Hospital of Integrated Chinese and Western Medicine | N |

all data were fairly complete. Moreover, for "selective reporting", all studies had a low risk of bias as they all employed standard analysis methods. Under the "other bias" category, only one study [49] showed an unclear risk of bias because patients' age was not reported at baseline.

## Primary outcome

**Objective response rate / Disease control rate.** Of the 26 selected studies, only 21 reported ORR/DCR for a total of 1370 patients. Furthermore, the meta-analysis revealed that TMPs+ VEGFR-TKIs co-treatment significantly increased the ORR by 14.3% (FEM; RR = 1.49; 95% CI: 1.31–1.69; $I^2$ = 0%; $p < 0.00001$; Fig 4), and DCR by 15.5% (FEM, RR = 1.23; 95% CI: 1.16–1.30; $I^2$ = 7%; $p < 0.00001$; Fig 5) compared to VEGRR-TKI mono-treatment.

**Sub-cohort assessment for tumor response.** Sub-cohort assessments for ORR and DCR were performed for the target drug regimen, treatment course, KPS score, and TMP form (Tables 3 and 4). The target drug regimen was categorized into sorafenib, apatinib, lenvatinib, and regorafenib. The sub-cohort analyses revealed that combining TMPs with lenvatinib can achieve better ORR (S1 Fig in S5 File). Furthermore, the TMP was given as a decoction, granule, pill, powder, bolus, and injection. The sub-cohort analysis revealed that the decoction

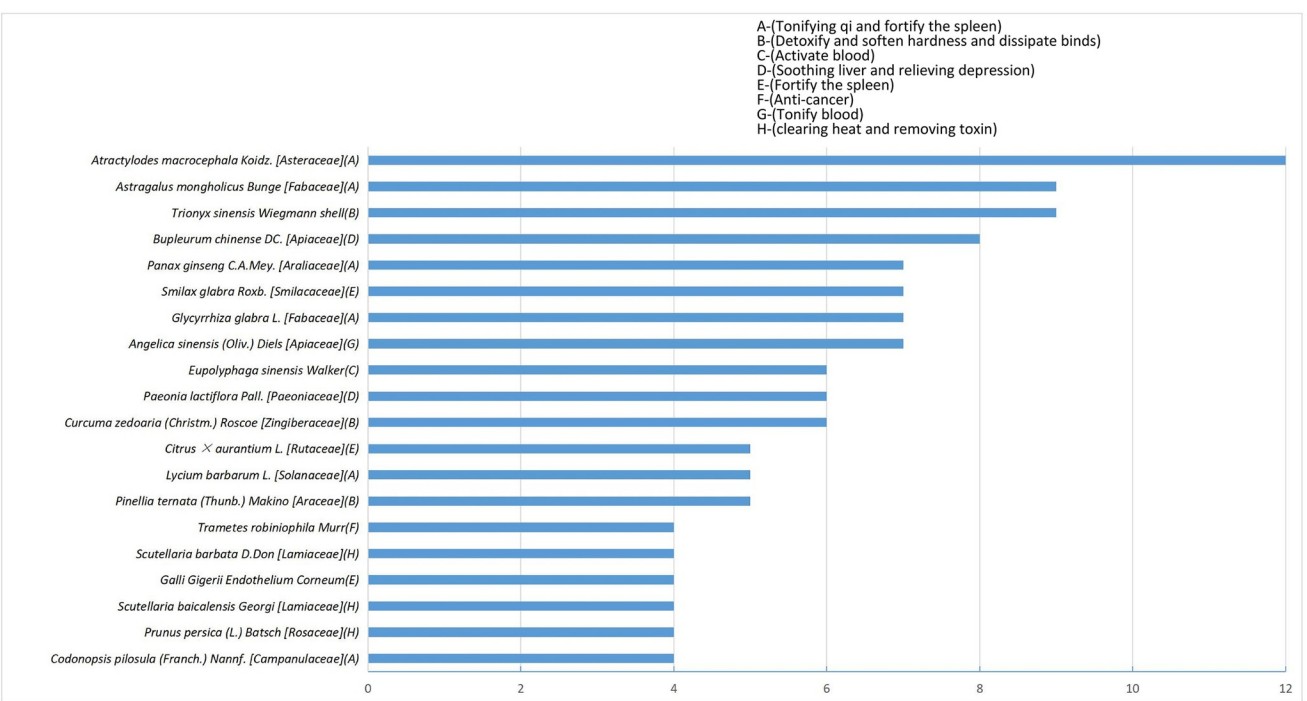

**Fig 2. Frequency and functional classification of the top 20 traditional medicines.**

form of TMPs can improve ORR better (S2 Fig in S5 File). Categorizing the treatment course [duration in months (M)] into < 2 and ≥ 2 M did not significantly affect the outcome (S3 Fig in S5 File). Furthermore, a subdivision of the KPS score to ≥ 60 and unclear did not significantly change the outcome (S4 Fig in S5 File). For DCR, the sorafenib sub-cohort indicated a better effect (S5 Fig in S5 File). Moreover, in DCR, the source of heterogeneity was the target drug regimen (sub-cohort difference analysis $p_h = 0.02$; $I_h^2 = 68.9$). In addition, the DCR of the remaining sub-cohorts had higher heterogeneity than the total pool (S6–S8 Figs in S5 File). Therefore, the sub-cohorts indicated no convincing results except for the targeted drug treatment schedule.

## Secondary outcomes

**One-year overall survival.** A total of 8 studies reported one-year OS was reported for 490 patients. The meta-analysis revealed that, similar to the effect on ORR, the addition of TMPs significantly increased one-year OS by 21.2% (FEM; RR = 1.49; 95% CI: 1.28–1.74; $I^2 = 23\%$; $p < 0.00001$) compare to VEGFR-TKI treatment alone (Fig 6).

**Quality of life.** The dichotomous KPS data were reported by 3 studies [43, 52, 61] for 171 patients. The meta-analysis presented no significant difference between TMPs + VEGFR-TKIs co-treatment and TKIs mono-treatment (FEM; RR = 1.12; 95% CI: 0.84–1.49; $I^2 = 0\%$; $p = 0.44$; S9 Fig in S5 File). Continuous KPS data were reported in 8 studies [47, 48, 58, 59, 61, 62, 64, 68] for 531 patients, but, the significant heterogeneity in these data prevented a meta-analysis. However, these eight studies indicated higher KPS scores in the co-treatment cohort compared to controls (S1 Table in S5 File).

**Alpha fetoprotein.** Continuous AFP data were documented in 13 studies comprising 899 patients (S2 Table in S5 File). Nine studies [44, 46, 48, 50, 52, 58, 60, 64, 67] reported data in

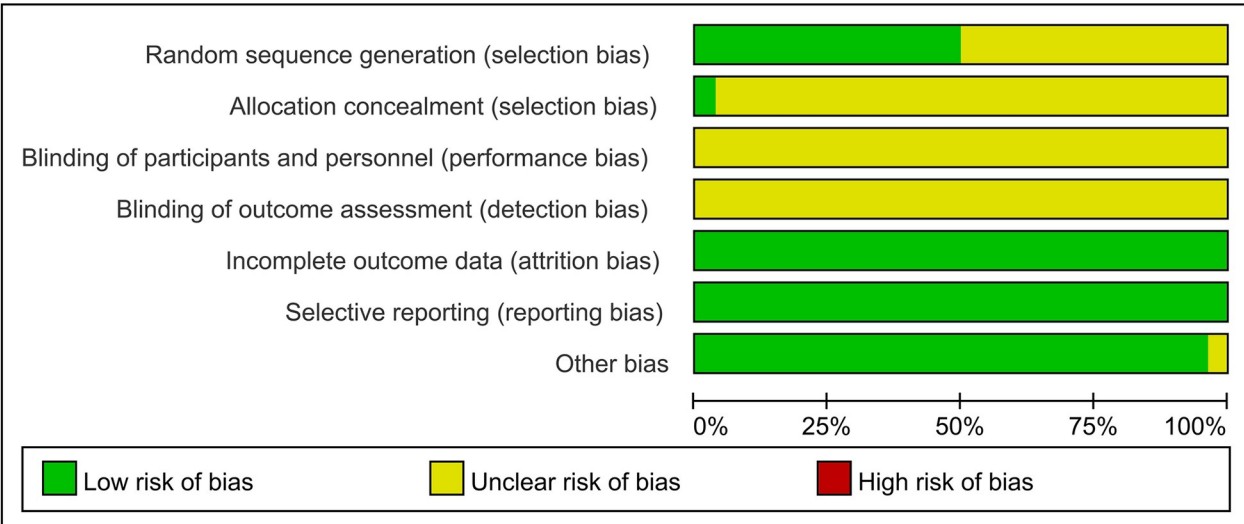

A

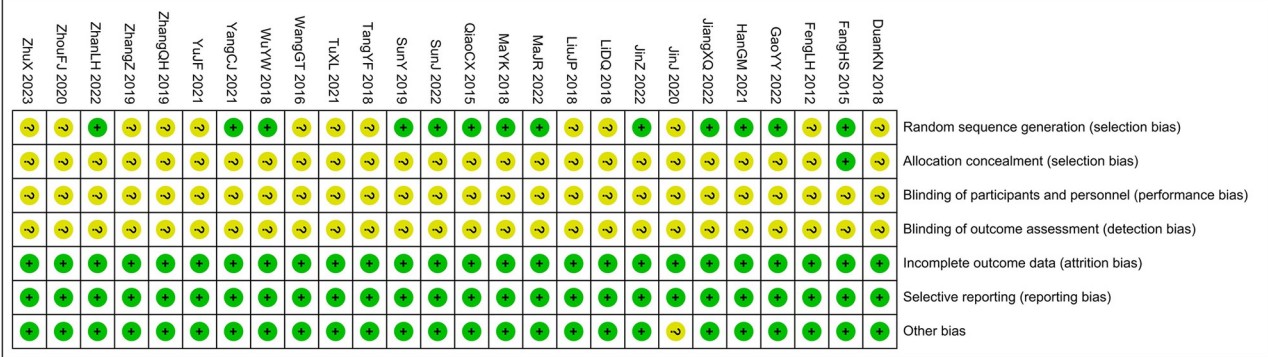

B

**Fig 3. Assessment of risk of bias in incorporated investigations.** (A) Graphical representation for the type of risk and their distribution. (B) Table summarizing the level of bias for all studies.

mean ± SD, while four [53, 61, 62, 68] reported the median and interquartile range. Unfortunately, the data were too heterogeneous to allow a meta-analysis. However, the consensus is that TMP addition can decrease AFP levels in patients compared with VEGFR-TKIs alone, as reported in ten studies. The remaining three investigations [61, 62, 64] indicated no major variations between the co-and mono-treatment. Only one study [54] identified AFP through dichotomous data, which also supports that combination treatment decreased AFP levels more than VEGFR-TKIs alone.

**Adverse drug reactions.**   The incidence of different ADRs is indicated in S3 Table within S5 File. Hypertension [44–46, 49, 52–54, 56, 58, 61, 62, 64, 65], hand-foot skin reactions [44, 46, 49, 52–54, 56, 61, 62, 64–67], and gastrointestinal reactions [44, 45, 50–53, 56, 58, 62, 64–67] were the most commonly reported ADR, reported in 14 studies. Followed by liver dysfunction, reported in 13 studies with 815 patients [5 dichotomous data [45, 49, 56, 61, 62], 8

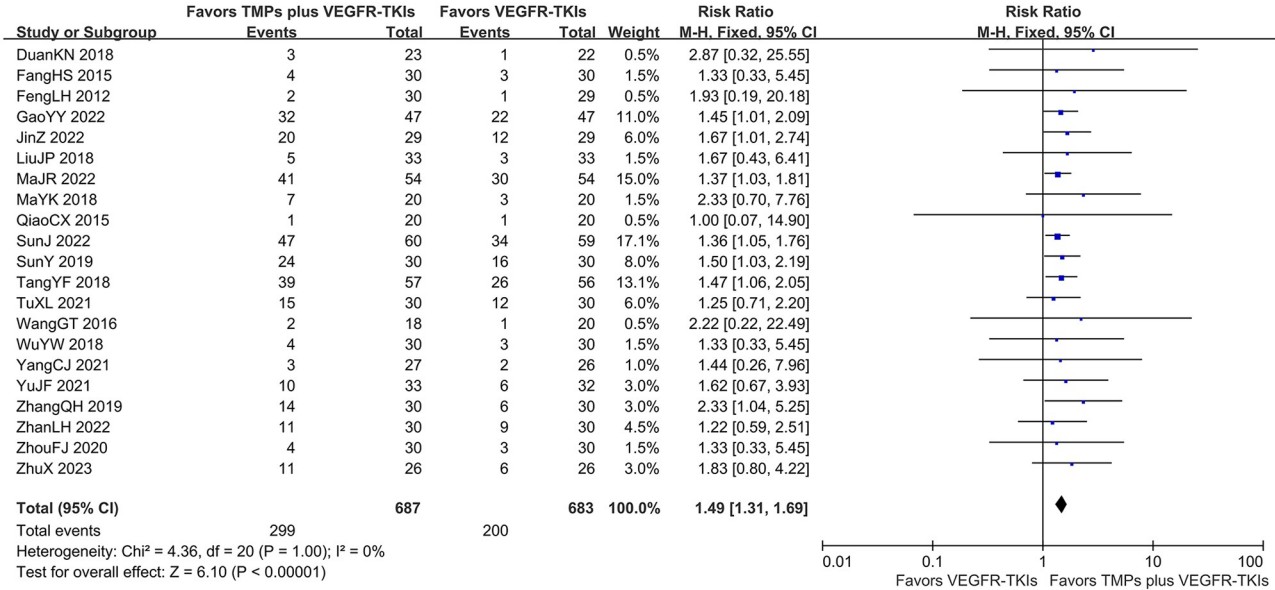

**Fig 4. Forest-plot/pooled RRs comparing the influence of TMPs + VEGFR-TKIs and VEGFR-TKIs on ORR.** Risk ratio represents the ratio of the studied outcome between the co-treatment group and the mono treatment group.

continuous data [44, 48, 51–53, 60, 64, 67]]. This was followed by myelosuppression [10 studies [44–46, 49, 54, 61, 62, 64, 66, 67]], fatigue [7 studies [44, 46, 52, 56, 64, 66, 67]], proteinuria [7 studies [46, 54, 61, 62, 64]] and rash [6 studies [44, 52, 58, 62, 65, 67]].

Meta-analysis revealed that compared to TKI mono-treatment, co-treatment reduce six manifestations from the eight evaluated ADRs, including liver dysfunction (dichotomous data: FEM, RR = 0.64, 95% CI: 0.45–0.91, 348 participants, $I^2 = 0\%$, $p = 0.01$), hypertension (FEM, RR = 0.66, 95% CI: 0.53–0.83, 1001 participants, $I^2 = 0\%$, $p = 0.0003$), hand-foot skin reactions (FEM, RR = 0.63, 95% CI: 0.49–0.80, 945 participants, $I^2 = 0\%$, $p = 0.0002$), gastrointestinal

**Table 3. Subgroup analysis of the ORR.**

| Subgroup | Number of trials | RR,95%CI | Z | P | Heterogeneity | |
|---|---|---|---|---|---|---|
| | | | | | $I^2$ | $P_h$ |
| Sorafenib | 12 | 1.52 [1.25, 1.85] | 4.14 | < 0.0001 | 0% | 1.00 |
| Lenvatinib | 3 | 1.53 [1.09, 2.16] | 2.44 | 0.01 | 0% | 0.68 |
| Apatinib | 5 | 1.47 [1.15, 1.89] | 3.06 | 0.002 | 0% | 0.93 |
| Regorafenib | 1 | 1.37 [1.03, 1.81] | 2.17 | 0.03 | Not applicable | Not applicable |
| 1M≤and<2M | 5 | 1.47 [1.04, 2.09] | 2.18 | 0.03 | 0% | 1.00 |
| ≥2M | 16 | 1.49 [1.30, 1.70] | 6.10 | < 0.00001 | 0% | 1.00 |
| Decoction | 11 | 1.52 [1.24, 1.87] | 3.69 | < 0.0001 | 0% | 1.00 |
| Granule | 6 | 1.50 [1.22, 1.83] | 3.93 | < 0.0001 | 0% | 0.84 |
| Powder | 1 | 1.33 [0.33, 5.45] | 0.40 | 0.69 | Not applicable | Not applicable |
| Injection | 1 | 1.36 [1.05, 1.76] | 2.35 | 0.02 | Not applicable | Not applicable |
| Pill | 1 | 1.93 [0.19, 20.18] | 0.55 | 0.58 | Not applicable | Not applicable |
| Bolus | 1 | 2.22 [0.22, 22.49] | 0.68 | 0.50 | Not applicable | Not applicable |
| ≥60 | 11 | 1.50 [1.22, 1.84] | 3.82 | 0.0001 | 0% | 0.99 |
| Unclear | 9 | 1.46 [1.23, 1.73] | 4.34 | < 0.0001 | 0% | 0.99 |

**Table 4. Subgroup analysis of the DCR.**

| Subgroup | Number of trials | RR,95%CI | Z | P | Heterogeneity | |
|---|---|---|---|---|---|---|
| | | | | | $I^2$ | $P_h$ |
| Sorafenib | 12 | 1.34 [1.21, 1.47] | 5.89 | < 0.0001 | 0% | 0.91 |
| Lenvatinib | 3 | 1.19 [1.03, 1.37] | 2.39 | 0.02 | 29% | 0.25 |
| Apatinib | 5 | 1.10 [1.00, 1.22] | 2.04 | 0.04 | 0% | 0.99 |
| Regorafenib | 1 | 1.09 [0.95, 1.24] | 1.22 | 0.22 | Not applicable | Not applicable |
| 1M≤and<2M | 5 | 1.29 [1.11, 1.51] | 3.26 | 0.001 | 44% | 0.13 |
| ≥2M | 16 | 1.21 [1.14, 1.29] | 5.91 | < 0.00001 | 0% | 0.47 |
| Decoction | 11 | 1.20 [1.11, 1.30] | 4.40 | < 0.0001 | 16% | 0.30 |
| Granule | 6 | 1.23 [1.10, 1.38] | 3.71 | 0.0002 | 9% | 0.36 |
| Powder | 1 | 1.36 [0.85, 2.17] | 1.27 | 0.20 | Not applicable | Not applicable |
| Injection | 1 | 1.20 [1.03, 1.39] | 2.33 | 0.02 | Not applicable | Not applicable |
| Pill | 1 | 1.48 [0.99, 2.22] | 0.06 | 1.91 | Not applicable | Not applicable |
| Bolus | 1 | 1.39 [0.92, 2.10] | 1.56 | 0.12 | Not applicable | Not applicable |
| ≥60 | 11 | 1.27 [1.14, 1.41] | 4.48 | < 0.00001 | 12% | 0.33 |
| Unclear | 9 | 1.20 [1.11, 1.29] | 4.78 | < 0.00001 | 20% | 0.27 |

reactions (REM, RR = 0.64, 95% CI: 0.45–0.92, 984 participants, $I^2$ = 50%, $p$ = 0.02), myelosuppression (FEM, RR = 0.63, 95% CI: 0.46–0.87, 599 participants, $I^2$ = 0%, $p$ = 0.005), and proteinuria (REM, RR = 0.43, 95% CI: 0.24–0.75, 418 participants, $I^2$ = 33%, $p$ = 0.003). The meta-analysis could not be performed for a continuous dataset for liver dysfunction (8 studies with 467 patients) because of the great heterogeneity. However, all eight studies showed lower levels of alanine aminotransferase in the co-treatment cohort compared to the control cohort post-therapy. In addition, the co-treatment cohort did not exhibit an increased incidence of rash

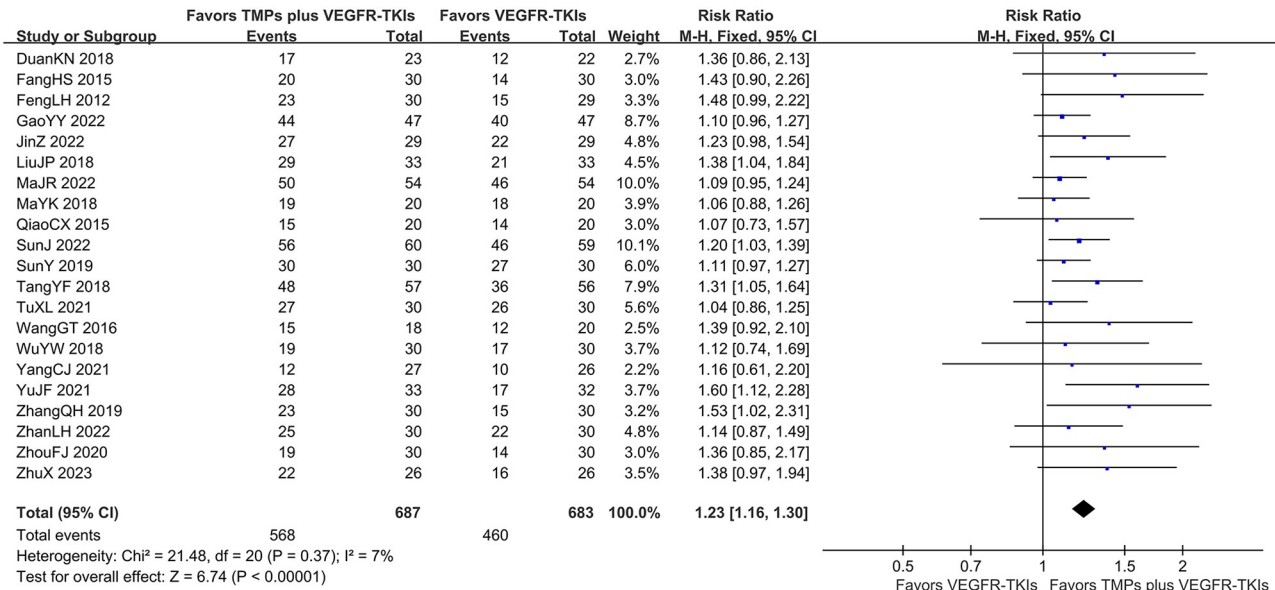

**Fig 5. Forest-plot/pooled RRs comparing the influence of TMPs + VEGFR-TKIs and VEGFR-TKIs on DCR.**

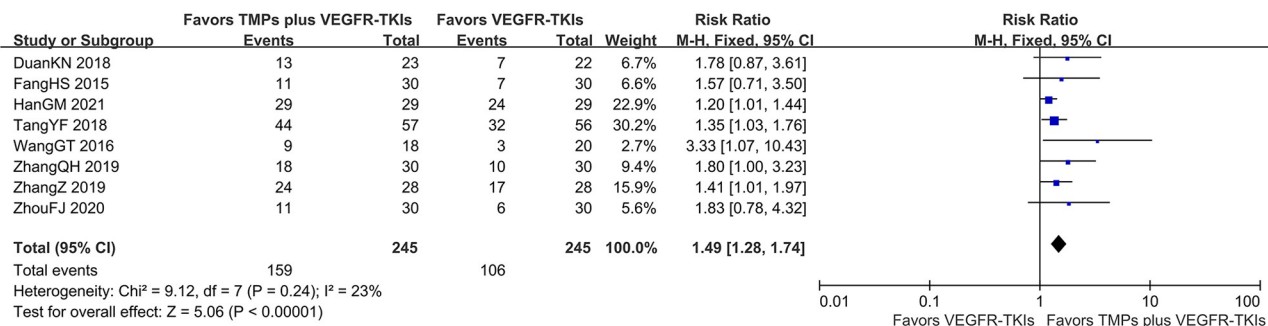

**Fig 6. Forest-plot/pooled RRs comparing the influence of TMPs + VEGFR-TKIs and VEGFR-TKIs on one-year OS.**

and fatigue (S10 Fig in S5 File). Meta-analysis outcomes for ADRs are depicted in S4 Table within S5 File and Fig 7.

**Influence of independent traditional medicine-based compounds on oral TMP cohort.** Oral TMPs included 105 different traditional medicine-derived compounds with a mean quantity of nine constituents for each TMP therapy. The peak frequency of traditional medicines use was as follows: *Atractylodes macrocephala* Koidz. [Asteraceae] (n = 12), *Astragalus mongholicus* Bunge [Fabaceae] (n = 9), *Trionyx sinensis* Wiegmann shell (n = 9), *Bupleurum chinense* DC. [Apiaceae] (n = 8), and *Panax ginseng* C.A.Mey. (n = 7). The sub-cohort analyses included 36 traditional medicines used in at least two studies. Of these, 10 traditional medicines indicated significant RRs for increasing ORR with low heterogeneity ($I^2 <$ 30%). These 10 were further assessed for combinations of pairs, triplets, or more with TMPs. Tables 5 and 6 present the sub-cohort analysis of ORR and DCR for these traditional medicines and their combinations.

*Sub-cohort analysis level 1. Single traditional medicines*. For ORR, the RRs of eight traditional medicines were higher than the total pool. *Panax ginseng* C.A.Mey. indicated the highest RR, followed by *Trametes robiniophila* Murr. (huaier), *Curcuma aromatica* Salisb. [Zingiberaceae], *Solanum nigrum* L. [Solanaceae], *Rehmannia glutinosa* (Gaertn.) DC. [Orobanchaceae], *Angelica sinensis* (Oliv.) Diels [Apiaceae], *Citrus × aurantium* L. [Rutaceae], *Prunus persica* (L.) Batsch [Rosaceae], *Paeonia lactiflora* Pall. [Paeoniaceae], and *Eupolyphaga sinensis* Walker.

For DCR, six traditional medicines had higher RRs than the total pool, with *Scutellaria barbata* D.Don [Lamiaceae] indicating the highest RR, followed by *Trametes robiniophila* Murr. (huaier), *Scolopendra subspinipes mutilans*, *Manis pentadactyla* Linnaeus, *Glycyrrhiza glabra* L. [Fabaceae], and *Prunus persica* (L.) Batsch [Rosaceae].

*Sub-cohort analysis level 2: Pairs of traditional medicines*. For ORR, nine pairs of traditional medicines had higher RRs than the total pool. The highest RR was for *Curcuma aromatica* Salisb. [Zingiberaceae] + *Angelica sinensis* (Oliv.) Diels [Apiaceae], followed by *Panax ginseng* C.A.Mey. + *Angelica sinensis* (Oliv.) Diels [Apiaceae], *Eupolyphaga sinensis* Walker + *Solanum nigrum* L. [Solanaceae], *Eupolyphaga sinensis* Walker + *Rehmannia glutinosa* (Gaertn.) DC. [Orobanchaceae], *Solanum nigrum* L. [Solanaceae] + *Rehmannia glutinosa* (Gaertn.) DC. [Orobanchaceae], *Curcuma aromatica* Salisb. [Zingiberaceae] + *Citrus × aurantium* L. [Rutaceae], *Citrus × aurantium* L. [Rutaceae] + *Pinellia ternata* (Thunb.) Makino [Araceae], *Angelica sinensis* (Oliv.) Diels [Apiaceae] + *Prunus persica* (L.) Batsch [Rosaceae], and *Curcuma aromatica* Salisb. [Zingiberaceae] + *Pinellia ternata* (Thunb.) Makino [Araceae]. Whereas, for

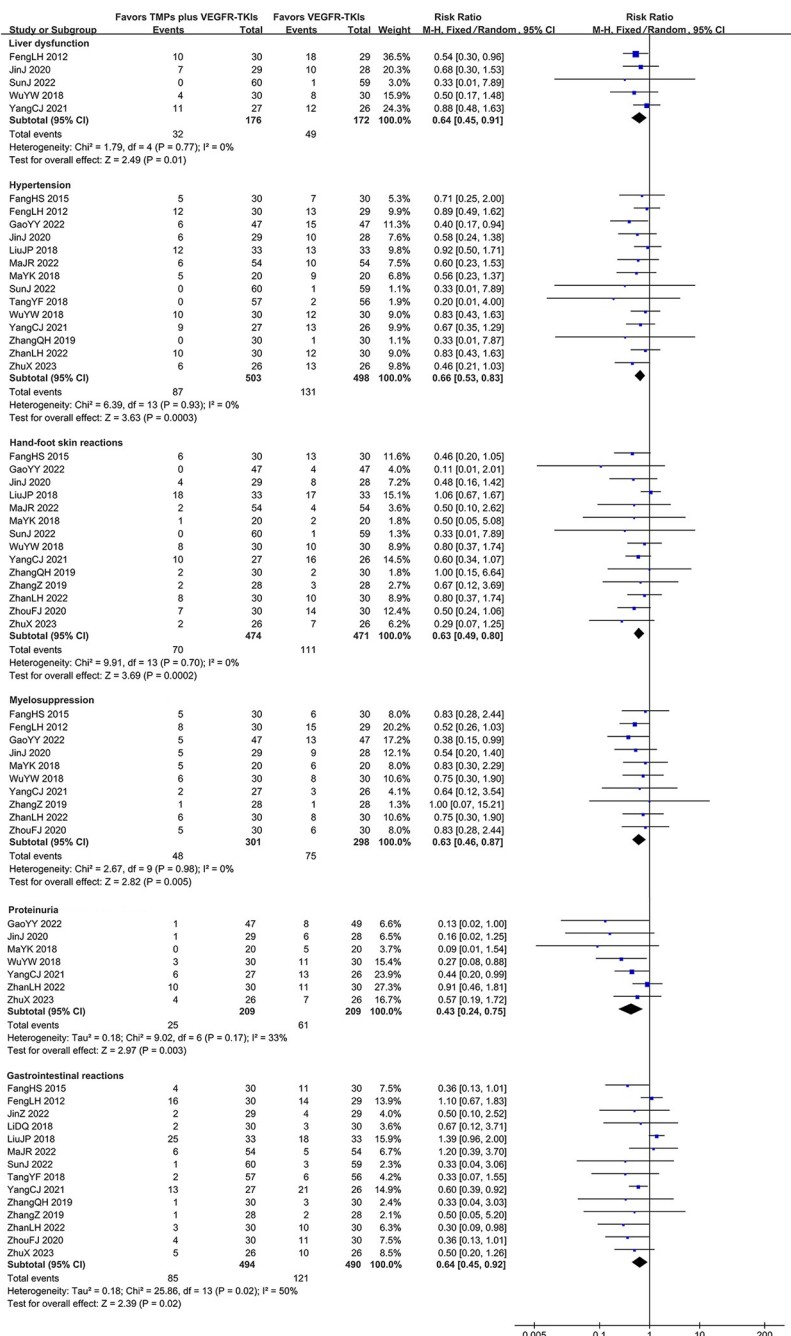

**Fig 7. Forest-plot/pooled RRs comparing the influence of TMPs + VEGFR-TKIs and VEGFR-TKIs on ADRs.**

DCR, only one pair of traditional medicines, *Scutellaria barbata* D.Don [Lamiaceae] + *Scolopendra subspinipes mutilans*, met the inclusion criteria.

*Sub-cohort analysis level 3*: *Triplets of traditional medicines*. It was observed that RRs for two triplets of traditional medicines were higher than the total pool. The highest RR was for *Eupolyphaga sinensis* Walker + *Solanum nigrum* L. [Solanaceae] + *Rehmannia glutinosa*

**Table 5.  Effects of specific TMPs on ORR for PLC: Single medicine and combinations.**

| Level | Traditional medicine | N studies | N participants | RR (95% CI) | $I^2$ |
|---|---|---|---|---|---|
| 1 | Panax ginseng C.A.Mey. | 6 | 316 | 1.65 [1.12, 2.44] | 0 |
| 1 | Trametes robiniophila Murr. | 2 | 173 | 1.63 [1.19, 2.24] | 10 |
| 1 | *Curcuma aromatica* Salisb. [Zingiberaceae] | 4 | 246 | 1.59 [1.15, 2.20] | 0 |
| 1 | *Solanum nigrum* L. [Solanaceae] | 2 | 105 | 1.58 [1.07, 2.34] | 0 |
| 1 | *Rehmannia glutinosa* (Gaertn.) DC. [Orobanchaceae] | 2 | 105 | 1.58 [1.07, 2.34] | 0 |
| 1 | *Angelica sinensis* (Oliv.) Diels [Apiaceae] | 5 | 390 | 1.56 [1.20, 2.02] | 0 |
| 1 | Citrus × aurantium L. [Rutaceae] | 3 | 187 | 1.55 [1.09, 2.20] | 0 |
| 1 | *Prunus persica* (L.) Batsch [Rosaceae] | 4 | 238 | 1.53 [1.13, 2.07] | 0 |
| 1 | *Paeonia lactiflora* Pall. [Paeoniaceae] | 5 | 276 | 1.51 [1.02, 2.23] | 0 |
| 1 | Eupolyphaga sinensis Walker | 5 | 290 | 1.50 [1.08, 2.08] | 0 |
| 2 | *Curcuma aromatica* Salisb. [Zingiberaceae] + *Angelica sinensis* (Oliv.) Diels [Apiaceae] | 2 | 92 | 2.00 [1.01, 3.97] | 0 |
| 2 | *Panax ginseng* C.A.Mey. + *Angelica sinensis* (Oliv.) Diels [Apiaceae] | 3 | 158 | 1.72 [1.10, 2.70] | 0 |
| 2 | Eupolyphaga sinensis Walker + Solanum nigrum L. [Solanaceae] | 2 | 105 | 1.58 [1.07, 2.34] | 0 |
| 2 | *Eupolyphaga sinensis* Walker + *Rehmannia glutinosa* (Gaertn.) DC. [Orobanchaceae] | 2 | 105 | 1.58 [1.07, 2.34] | 0 |
| 2 | *Solanum nigrum* L. [Solanaceae] + *Rehmannia glutinosa* (Gaertn.) DC. [Orobanchaceae] | 2 | 105 | 1.58 [1.07, 2.34] | 0 |
| 2 | *Curcuma aromatica* Salisb. [Zingiberaceae] + *Citrus × aurantium* L. [Rutaceae] | 2 | 134 | 1.56 [1.09, 2.22] | 0 |
| 2 | *Citrus × aurantium* L. [Rutaceae] + *Pinellia ternata* (Thunb.) Makino [Araceae] | 3 | 187 | 1.55 [1.09, 2.20] | 0 |
| 2 | *Curcuma aromatica* Salisb. [Zingiberaceae] + *Pinellia ternata* (Thunb.) Makino [Araceae] | 3 | 194 | 1.54 [1.08, 2.18] | 0 |
| 2 | *Angelica sinensis* (Oliv.) Diels [Apiaceae] + *Prunus persica* (L.) Batsch [Rosaceae] | 4 | 238 | 1.53 [1.13, 2.07] | 0 |
| 3 | *Eupolyphaga sinensis* Walker + *Solanum nigrum* L. [Solanaceae] + *Rehmannia glutinosa* (Gaertn.) DC. [Orobanchaceae] | 2 | 105 | 1.58 [1.07, 2.34] | 0 |
| 3 | *Curcuma aromatica* Salisb. [Zingiberaceae] + *Pinellia ternata* (Thunb.) Makino [Araceae] + *Citrus × aurantium* L. [Rutaceae] | 2 | 134 | 1.56 [1.09, 2.22] | 0 |

**Table 6.  Effects of specific TMPs on DCR for PLC: Single herb and combinations.**

| Level | Traditional medicine | N studies | N participants | RR (95% CI) | $I^2$ |
|---|---|---|---|---|---|
| 1 | Scutellaria barbata D.Don [Lamiaceae] | 3 | 171 | 1.39 [1.11,1.73] | 0 |
| 1 | Trametes robiniophila Murr. | 2 | 173 | 1.38 [1.13,1.68] | 0 |
| 1 | Scolopendra subspinipes mutilans | 2 | 111 | 1.37 [1.07,1.75] | 0 |
| 1 | Manis pentadactyla Linnaeus | 2 | 126 | 1.37 [1.06,1.77] | 0 |
| 1 | Glycyrrhiza glabra L. [Fabaceae] | 6 | 398 | 1.25 [1.10, 1.41] | 29 |
| 1 | *Prunus persica* (L.) Batsch [Rosaceae] | 4 | 238 | 1.25 [1.07,1.44] | 11 |
| 2 | Scutellaria barbata D.Don [Lamiaceae] + Scolopendra subspinipes mutilans | 2 | 111 | 1.37 [1.07,1.75] | 0 |

(Gaertn.) DC. [Orobanchaceae], followed by *Curcuma aromatica* Salisb. [Zingiberaceae] + *Pinellia ternata* (Thunb.) Makino [Araceae] + *Citrus × aurantium* L. [Rutaceae]. No triplet combinations were found for DCR.

## Publication bias

Funnel plots comparing TMPs + VEGFR-TKIs co-treatment and VEGFR-TKIs mono-treatment for ORR, DCR, hypertension, hand-foot skin reactions, and gastrointestinal reactions showed symmetric distribution on both sides (Fig 8), indicating no significant publication bias. These data were validated by Begg's test, which indicated insignificant publication bias for ORR (*p = 0.3323*), hand-foot skin reactions (*p = 0.3506*), and gastrointestinal reactions (*p = 0.9562*). However, potential publication bias was indicated for DCR (*p = 0.0852*), and hypertension (*p = 0.0546*). Following the results of the funnel plot and Begg's test, we further conducted Egger's tests for hypertension (*p = 0.1405, intercept = 0.0526*) and hand-foot skin reactions (*p = 0.0952, intercept = 0.0180*) to enhance the rigor of our assessment. A significance level of *p < 0.1* is recommended for detecting publication bias in both Begg's and Egger's tests [69]. After a comprehensive evaluation, we concluded that there is potential publication bias in the results for DCR, hypertension, and hand-foot skin reactions.

## Sensitivity analyses

Since both random and fixed effect models were employed in this meta-analysis, determining whether switching between them would significantly alter the outcome was important. Sensitivity analyses showed that the acquired results were stable when switching between these modeling methods.

## Quality of evidence

The quality of evidence was assessed using the GRADE criteria. Data on ORR, one-year OS, and myelosuppression were rated moderate, while that on DCR, hypertension, hand-foot skin reactions, liver dysfunction, gastrointestinal reactions, and proteinuria were rated low, whereas data on the quality-of-life was rated very low (Table 7).

## Discussion

### Predominant review outcomes

In this review, the data included 26 studies indicated was mostly uncertain, had low risk of bias, and was of moderate quality. The meta-analysis revealed that co-therapy of TMPs + VEGFR-TKIs considerably increased the ORR and DCR compared to VEGFR-TKIs alone. This might be because TMP can inhibit HCC cell proliferation and migration [29], suppress

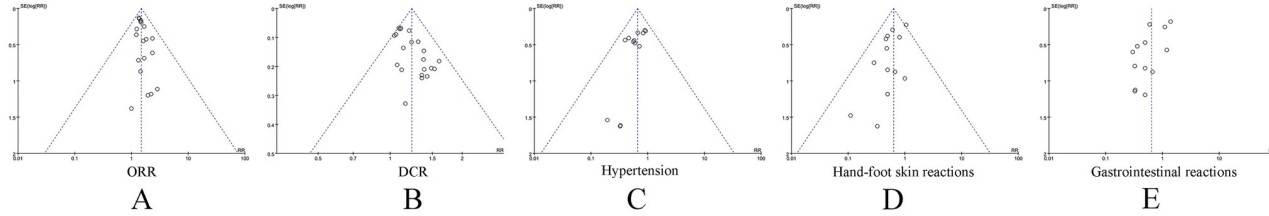

**Fig 8. Assessment of publication bias.** Funnel plots comparing TMPs + VEGFR-TKIs with VEGFR-TKIs mono-treatment for (A) ORR (21 trials), (B) DCR (21 trials), (C) hypertension (14 trials), (D) hand-foot skin reactions (14 trials), and (E) gastrointestinal reactions (14 trials).

**Table 7. Summary of finding table of TMPs plus VEGFR-TKIs compared to VEGFR-TKIs alone for PLC.**

**TMPs plus VEGFR-TKIs compared to VEGFR-TKIs alone for PLC**

**Patient or population:** Middle-advanced Primary Liver Cancer
**Setting:** Outpatient department/Inpatient department
**Intervention:** TMPs plus VEGFR-TKIs
**Comparison:** VEGFR-TKIs

| Outcome № of participants (studies) | Relative effect (95% CI) | Anticipated absolute effects (95% CI) | | | Certainty |
|---|---|---|---|---|---|
| | | VEGFR-TKIs alone | TMPs plus VEGFR-TKIs | Difference | |
| ORR № of participants: 1370 (21 RCTs) | RR 1.49 (1.31 to 1.69) | 29.3% | 43.6% (38.4 to 49.5) | 14.3% more (9.1 more to 20.2 more) | ⊕⊕⊕◯ Moderate[a] |
| DCR № of participants: 1370 (21 RCTs) | RR 1.23 (1.16 to 1.30) | 67.3% | 82.8% (78.1 to 87.6) | 15.5% more (10.8 more to 20.2 more) | ⊕⊕◯◯ Low[a,b] |
| One-year OS № of participants: 490 (8 RCTs) | RR 1.49 (1.28 to 1.74) | 43.3% | 64.5% (55.4 to 75.3) | 21.2% more (12.1 more to 32 more) | ⊕⊕⊕◯ Moderate[a] |
| KPS (dichotomous data) № of participants: 171 (3 RCTs) | RR 1.12 (0.84 to 1.49) | 41.2% | 46.1% (34.6 to 61.4) | 4.9% more (6.6 fewer to 20.2 more) | ⊕◯◯◯ Very low[a,c,d] |
| Liver dysfunction № of participants: 348 (5 RCTs) | RR 0.64 (0.45 to 0.91) | 28.5% | 18.2% (12.8 to 25.9) | 10.3% fewer (15.7 fewer to 2.6 fewer) | ⊕⊕◯◯ Low[a,e] |
| Proteinuria № of participants: 418 (7 RCTs) | RR 0.43 (0.24 to 0.75) | 29.2% | 12.6% (7 to 21.9) | 16.6% fewer (22.2 fewer to 7.3 fewer) | ⊕⊕◯◯ Low[a,f] |
| Hypertension № of participants: 1001 (14 RCTs) | RR 0.66 (0.53 to 0.83) | 26.3% | 17.4% (13.9 to 21.8) | 8.9% fewer (12.4 fewer to 4.5 fewer) | ⊕⊕◯◯ Low[a,b] |
| Hand-foot skin reactions № of participants: 945 (14 RCTs) | RR 0.63 (0.49 to 0.80) | 23.6% | 14.8% (11.5 to 18.9) | 8.7% fewer (12 fewer to 4.7 fewer) | ⊕⊕◯◯ Low[a,b] |
| Myelosuppression № of participants: 599 (10 RCTs) | RR 0.63 (0.46 to 0.87) | 25.2% | 15.9% (11.6 to 21.9) | 9.3% fewer (13.6 fewer to 3.3 fewer) | ⊕⊕⊕◯ Moderate[a] |
| Gastrointestinal reactions № of participants: 984 (14 RCTs) | RR 0.64 (0.45 to 0.92) | 24.7% | 15.8% (11.1 to 22.7) | 8.9% fewer (13.6 fewer to 2 fewer) | ⊕⊕◯◯ Low[a,f] |

*The risk in the intervention group (and its 95% confidence interval) is based on the assumed risk in the comparison group and the relative effect of the intervention (and its 95% CI).
CI: confidence interval; RR: risk ratio

GRADE Working Group grades of evidence
**High certainty:** we are very confident that the true effect lies close to that of the estimate of the effect.
**Moderate certainty:** we are moderately confident in the effect estimate: the true effect is likely to be close to the estimate of the effect, but there is a possibility that it is substantially different.
**Low certainty:** our confidence in the effect estimate is limited: the true effect may be substantially different from the estimate of the effect.
**Very low certainty:** we have very little confidence in the effect estimate: the true effect is likely to be substantially different from the estimate of effect.

Explanations
[a]. There were serious limitations of methodological quality among trials according to the risk of bias assessment.

[b]. There was potential publication bias.

[c]. Too small simple size.

[d]. There was no difference between the experience group and control group according to the *p-value*.

[e]. There was no significant difference between the experience group and control group according to the *p-value*.

[f]. There was significant statistical heterogeneity among trials according to $I^2$

HCC cell cycle [70], and reduce angiogenesis [71]. Pharmacological research has indicated that nobiletin, an extract from *Citrus × aurantium*, can suppress HCC cell proliferation and apoptosis by downregulating Bcl-2 and COX-2, as well as upregulating Bax and caspase-3 [26]. Furthermore, catalpol, an *R. glutinosa* extract, can synergistically increase the antitumor effects of regorafenib in HepG2 and HUH-7 HCC cell lines by inhibiting VEGF/VEGFR2, PI3K/Akt/mTOR, and NF-κB signaling pathways [72].

It has been revealed that in terms of OS, lenvatinib is comparable to sorafenib clinically, but has improved ORR [9]. This investigation revealed that TMPs + lenvatinib had better ORR than TMP + other VEGFR-TKIs. Regarding DCR, this investigation revealed that TMPs + sorafenib co-treatment was better than TMP + other VEGFR-TKIs, implying that TMPs may synergistically potentiate the anti-tumor effects of sorafenib/ lenvatinib against middle-advanced PLC. However, this result may be biased because there were few available studies on lenvatinib and regorafenib for metastasis. Further investigations are required to determine the effect of co-treatment of TMPs with different VEGFR-TKI regimens against various tumors.

In addition to improving ORR and DCR, the meta-analysis also revealed that TMPs + VEGFR-TKIs had a better effect on one-year OS than VEGFR-TKIs mono-treatment. Furthermore, all the studies that reported one-year OS included TMPs + sorafenib co-treatment. However, this result only supports TMPs + sorafenib co-treatment for improved one-year survival rate and cannot be extended to all other VEGFR-TKIs. This synergistic effect between TMPs and sorafenib may be due to the anti-tumor effect of TMPs [29], or its ability to reverse resistance to targeted drugs [70].

Although co-treatment of TMPs + VEGFR-TKIs significantly improved the primary outcome and one-year OS, the quality of life was not substantially altered based on dichotomous data, which might be because of the low quality of the evidence. The continuous data showed more encouraging trends with the co-treatment increasing the quality of life. This may be closely related to the lower incidence of ADRs within the TMPs + VEGFR-TKIs cohort.

Increased AFP level is a risk factor for PLC [73], and it is essentially associated with diagnosis, prognosis, and PLC monitoring [74]. Therefore, for this outcome, due to dataset heterogeneity, only qualitative analysis was performed. Here, ten trials showed that TMPs + VEGFR-TKIs reduced AFP, while three indicated no advantage for the co-therapy. In a retrospective cohort study, sorafenib combined with Fufang Banmao capsules, a type of TMP including *C. aromatica* and *S. barbata*, was found to decrease AFP levels in HCC patients [75]. Furthermore, icaritin, an extract from *Epimedium sagittatum* (Siebold & Zucc.) Maxim. [Berberidaceae], was observed to inhibit HCC metastasis by downregulating *AFP* gene expression [76]. This indicates that TMPs have the potential to reduce AFP and need further exploration.

The meta-analysis of moderate quality evidence suggested that, compared to control treatments, co-treatment with TMPs and VEGFR-TKIs may reduce the incidence of myelosuppression as an adverse drug reaction (ADR). Moreover, low quality evidence suggested that the co-treatment group had a lower incidence of liver dysfunction, hand-foot skin reactions, hypertension, gastrointestinal reactions, and proteinuria compared to the control group. The reduced ADRs might be related to the multi-component, -target, and -channel functions of TMPs. It has been revealed that some TMPs [77] and traditional medicines such as *Astragalus* [78] can treat proteinuria, some TMPs can control hypertension [79], and ginsenoside Rg3 can reduce myelosuppression [80]. Moreover, this study also found that TMPs + VEGFR-TKIs co-treatment does not increase the incidence of some common ADRs, such as rash and fatigue, compared with the control treatment. Therefore, the results of this investigation support that TMPs + VEGFR-TKIs co-therapy may be safe for middle-advanced PLC treatment.

## Implications for clinical practice

This review showed that TMPs combined that VEGFR-TKIs have better effectiveness and safety profile for treating patients with middle-advanced PLC than VEGFR-TKIs alone. This improved efficacy might be because of some traditional medicine's ability to exert anti-tumor effects through multiple targets and pathways. Moreover, some traditional medicines can reverse multi-drug resistance and synergistically increase the anti-tumor effects of VEGFR-T-KIs. Here, it was identified that the main TMP's mechanisms included heat clearance, toxin removal, tonifying, and blood activation, liver soothing, depression relief, qi tonifying, and spleen fortifying. The most frequently used TMPs and traditional medicine in this review were Huaier granule and *A. macrocephala*, respectively. Based on this meta-analysis, the following traditional medicines are recommended for middle-advanced PLC treatment when combined with VEGFR-TKIs: *P. ginseng*, *T. robiniophila* Murr., *S. nigrum*, *R. glutinosa*, *A. sinensis*, *Citrus × aurantium*, *C. aromatica*, *P. persica*, *E. sinensis* walker, *S. barbata*, *S. s. mutilans*, *M. pentadactyla* Linnaeus, and *G. glabra* as they have a significant effect on ORR/DCR. Additionally, the following combinations of traditional medicines are recommended for use in conjunction with VEGFR-TKIs: *E. sinensis* Walker + *S. nigrum* + *R. glutinosa*, *C. aromatica* + *P. ternate* + *Citrus × aurantium*, and *S. barbata* + *S. s. mutilans*. The extracts and compounds of these 12 traditional medicines have potential *in vivo* and *in vitro* therapeutic effects on PLC. Their pharmacological actions and mechanisms are summarized in S5 Table within S5 File.

## Advantages and limitations of the review

In recent years, several systematic reviews of TMPs and systematic treatments of malignant tumors have been published, which suggest the potential anti-tumor effects of TMPs. Although previous review topics are similar, this review is different in several ways. 1) Methodologically, this research assessed evidence quality using GRADE criteria, a recognized standard procedure. 2) The sources of heterogeneity with sub-cohort analysis were also assessed. 3) The PRISMA guidelines were followed for meta-analysis studies [81]. Wang *et al*. reviewed the effect of TMPs + TACE co-treatment compared to TACE mono-treatment in PLC patients and revealed that the co-treatment could improve the effective rate and reduce serum VEGF levels [82]. However, this study differs from the current acquired data due to the treatment evaluation. Xun *et al*. reviewed 12 studies to evaluate the effect of TMPs + sorafenib in PLC patients and included all types of VEGFR-TKIs [83]. Although the treatment strategies are similar, this study specifically focused on the middle and advanced stages of PLC rather than the whole PLC. The results of this and Xun's study were consistent for tumor response, quality-of-life, and a part of ADRs. However, in this study, gastrointestinal reactions, rash, and fatigue, the co-treatment cohort displayed no advantage over the control cohort. These differences might be because here, each type of ADR was evaluated separately. This work may be the first to probe the effectiveness and safety profiles for TMPs when in combination with VEGFR-TKIs as middle-advanced PLC therapy.

However, there are several limitations in this study. 1) Only English and Chinese language databases were surveyed, therefore, omissions may exist. All the included studies were sourced from China because of the main location for Chinese medicine research, which may have regional and ethnic influences. Further research is thus required to extend the current findings to other populations and regions. 2) Only 13 studies [44, 46–48, 50, 53–57, 61, 62, 64] reported random sequence generation, and none followed the blinding method, which may have introduced potential selection and detection biases. Therefore, the GRADE rating for the evidence quality of these data might be low or very low. However, the sensitivity analysis showed that the outcomes were reliable. The data will be constantly updated with the publication of new

high-quality studies. 3) Only 8 studies [43, 44, 47, 58, 60, 65–67] reported the long-term survival in PLC patients, thus, warrants further research. 4) Although all studies reported quality control, only 8 studies [45, 50, 51, 53, 56, 58, 62, 65] included chemical analysis; the active ingredients of TMPs in other studies were unclear and required clarification from the drug manufacturers or research studies, which may cause bias. In addition, sensitivity and sub-cohort analyses were carried out to mitigate the effect of heterogeneity and the resulting bias for specific traditional medicine within TMPs. 5) The majority of the studies reviewed within this investigation did not adhere stringently to reporting standards for CONSORT Extension for Chinese Herbal Medicine Formulas [84]. 6) Using funnel plots, Begg's test, and Egger's test, we identified potential publication bias in DCR, hypertension, and hand-foot skin reactions. This may be attributable to publication bias favoring positive results, wherein studies with significant findings are more likely to be published. Additionally, some of the included studies had small sample sizes, which, if they happened to demonstrate larger effects, may have been more readily published, even though these effects may not be broadly representative. This can lead to potential publication bias. Furthermore, the literature search was restricted to studies published in English and Chinese, which may have contributed to bias as well. We recommend that future research efforts include pre-registered studies and encourage both researchers and journals to publish studies with non-significant findings to mitigate the impact of publication bias.

## Areas for future research

Several areas in this field warrant research in the future. First, it will be worth exploring the mechanism of traditional medicines for the synergistic potentiation of VEGFR-TKIs. Second, the effect of traditional medicine in combinations such as *C. aromatica* + *P. ternata* + *Citrus × aurantium* should be evaluated. Furthermore, the mechanism of action of traditional medicines can be assessed by isolating and purifying the monomers to identify their active ingredients, such as RG-3 extracted from *P. ginseng* C.A.Mey. [Araliaceae]. Third, since only eight trials reported one-year OS, more studies with long-term follow-up will significantly improve our understanding of the prolonged benefit of TMPs and their combinatorial uses. Fourth, ADR incidence was commonly reported; however, for therapeutic decision making more information should be recorded. Therefore, it is recommended to use standardized evaluating tools including NCI-CTCAE (National Cancer Institute-Common Toxicity Criteria for Adverse Events) for enhanced identification of ADR extent. Fifth, for clinical application and comprehensive understanding of this therapy, well-designed and high-quality clinical investigations are required. Thus, researchers should follow established good practices including the CONSORT Extension for Chinese Herbal Medicine Formulas guideline [84] for validating the effectiveness and safety profile of combinatory therapies of TMPs and VEGFR-TKIs for PLC. Sixth, because of the unclear TMP composition, the investigation of clinical outcomes is restricted. Future studies must preferably include quality control and chemical analysis of TMPs.

## Conclusions

This investigation indicated that TMPs + VEGFR-TKIs co-treatment has enhanced efficacy and reduced risk of ADRs, for middle-advance PLC patients. Furthermore, only the "TMPs + sorafenib" combination improved the one-year survival rate in long-term prognosis. In addition, TMPs + lenvatinib co-treatment has better ORR than the combination of TMPs with other VEGFR-TKIs. When it comes to DCR, TMPs + sorafenib co-treatment was better. Moreover, it was indicated that TMPs can reduce AFP and need further exploration. Based on

the meta-analysis, it was recommended that for middle-advanced PLC treatment, the following traditional medicines should be combined with VEGFR-TKIs: *P. ginseng*, *T. robiniophila* Murr., *S. nigrum*, *R. glutinosa*, *A. sinensis*, *Citrus × aurantium*, *C. aromatica*, *P. persica*, *E. sinensis* walker, *S. barbata*, *S. s. mutilans*, *M. pentadactyla* Linnaeus, and *G. glabra*, as they have a significant effect on ORR/DCR. Additionally, following combinations of traditional medicines with VEGFR-TKIs were also recommended: *E. sinensis* Walker + *S. nigrum* + *R. glutinosa*; *C. aromatica* + *P. ternate* + *Citrus × aurantium*; *S. barbata* + *S. s. mutilans*. However, further verification is required with well-designed and high-quality clinical trials adhering to CONSORT Extension for Chinese Herbal Medicine Formulas guidelines.

## Supporting information

**S1 File. PRISMA checklist.** The PRISMA checklist is available in S1 File.
(DOCX)

**S2 File. Detailed search strategy.** The detailed search strategy is available in S2 File.
(DOCX)

**S3 File. Literature list.**
(XLSX)

**S4 File. Risk of bias assessment.**
(XLSX)

**S5 File. Attached figures and table.** The S1–S10 Figs and S1-S5 Tables are available in S5 File.
(DOCX)

## Acknowledgments

The authors would like to thank all the reviewers who participated in the review and MJEditor (www.mjeditor.com) for its linguistic assistance during the preparation of this manuscript.

## Author Contributions

**Conceptualization:** Hui-Bo Yu, Hong-Gang Zheng.

**Data curation:** Hui-Bo Yu, Jia-Qi Hu, Bao-Jin Han, Shun-Tai Chen.

**Formal analysis:** Hui-Bo Yu, Jia-Qi Hu, Bao-Jin Han.

**Investigation:** Shun-Tai Chen.

**Methodology:** Yan-Yuan Du, Shun-Tai Chen, Xin Chen, Hong-Tai Xiong, Jin Gao.

**Resources:** Yan-Yuan Du.

**Software:** Hui-Bo Yu.

**Writing – original draft:** Hui-Bo Yu, Bao-Jin Han.

**Writing – review & editing:** Hong-Gang Zheng.

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
