## [Decision Letter · Decision Letter 0]

5 Feb 2024

PONE-D-23-16652Combinatorial Treatment with Traditional Medicinal Preparations and VEGFR-Tyrosine Kinase Inhibitors for Middle-Advanced Primary Liver Cancer: A Systematic Review and Meta-AnalysisPLOS ONE

Dear Dr. Gang Zhenga,

Thank you for submitting your manuscript to PLOS ONE. After careful consideration, we feel that it has merit but does not fully meet PLOS ONE’s publication criteria as it currently stands. Therefore, we invite you to submit a revised version of the manuscript that addresses the points raised during the review process.

Please submit your revised manuscript by Mar 21 2024 11:59PM. If you will need more time than this to complete your revisions, please reply to this message or contact the journal office at plosone@plos.org. Please include the following items when submitting your revised manuscript:A rebuttal letter that responds to each point raised by the academic editor and reviewer(s). You should upload this letter as a separate file labeled 'Response to Reviewers'.A marked-up copy of your manuscript that highlights changes made to the original version. You should upload this as a separate file labeled 'Revised Manuscript with Track Changes'.An unmarked version of your revised paper without tracked changes. You should upload this as a separate file labeled 'Manuscript'.

We look forward to receiving your revised manuscript.

Kind regards,

Hazem Osama Radwan Elkady

Academic Editor

PLOS ONE

Journal Requirements:

2. Please remove any discussion of unvalidated diagnoses or methods, specifically in the penultimate paragraph of the introduction section.

“This study was supported by the National Natural Science Foundation of China (Grant No. 82174463), Scientific and Technological Innovation Project of China Academy of Chinese Medical Sciences (No. CI2021A01804), Innovation Team and Talents Cultivation Program of National Administration of Traditional Chinese Medicine (No. ZYYCXTD-C-202205).”

Reviewers' comments:

Reviewer's Responses to Questions

**Comments to the Author**

1. Is the manuscript technically sound, and do the data support the conclusions?

Reviewer #1: Partly

Reviewer #2: Partly

Reviewer #3: Yes

2. Has the statistical analysis been performed appropriately and rigorously? 

Reviewer #1: Yes

Reviewer #2: I Don't Know

Reviewer #3: Yes

3. Have the authors made all data underlying the findings in their manuscript fully available?

Reviewer #1: Yes

Reviewer #2: Yes

Reviewer #3: Yes

4. Is the manuscript presented in an intelligible fashion and written in standard English?

Reviewer #1: No

Reviewer #2: Yes

Reviewer #3: Yes

5. Review Comments to the Author

Reviewer #1: The study assesses a current, timely topic in HCC.

We recommend some changes:

- We believe this article is suitable for publication in the journal although major revisions are needed. The main strengths of this paper are that it addresses an interesting and very timely question and provides a clear answer, with some limitations. Certainly, the authors should better highlight the limitations of the current paper.

- A linguistic revision is needed.

- The background of the changing scenario of medical treatment in HCC should be better discussed, and some recent papers regarding this topic should be included ( PMID: 32772560; PMID: 36633661; PMID: 34798793; PMID: 35031442 ).

Major changes are necessary.

Reviewer #2: The efforts exerted in this current work is so appreciated. Some points need to be addressed. So, a minor revision may be required to improve the manuscript:

1- Some typos and grammatical errors ought to be corrected through the whole manuscript.

2- The article has too many abbreviations, which might cause confusion to readers. I recommend to reduce the number of abbreviated terms within the manuscript.

3- The introduction suffers from poor discussion about Cancer/VEGFR relationship and VEGFR-TK inhibitors. Please improve.

4- “Meta-analysis was performed through two scientists (HBY and JQH) independently”. I do no prefer to cite the authors’ names here. It is much better to talk about it generally.

5- It is not recommended to use abbreviations in heading sections (e.g QoL, AFP, ADRs, etc…)

6- The conclusion part has to be improved. It should include deep insights.

7- All figures’ resolution have to be improved.

Reviewer #3: In the current systematic review, the authors evaluated the therapeutic efficacy and safety of Traditional medicine preparations (TMPs) combined with vascular endothelial growth factor receptor (VEGFR)-associated multi-targeted tyrosine kinase inhibitors (TKIs) for middle-advanced primary liver cancer (PLC). The analysis revealed that combinatorial treatment with TMPs and VEGFR-TKIs may exhibit a higher effectiveness and safety profile within middle-advanced PLC therapy.

I recommend publishing this work in PlosOne after addressing the following minor points.

- The rationale for this review and its aims should be more clear for the readers.

- I wonder if the authors could provide more recent studies from 2023, as I can see there is no any study from 2023.

- Also, it is worth mentioning that there is a limited number of studies from 2022.

- The manuscript contains some errors and inaccuracies in terms of typography. I kindly request you to review it carefully and make the necessary revisions.

6. PLOS authors have the option to publish the peer review history of their article (what does this mean?). If published, this will include your full peer review and any attached files.

Reviewer #1: No

Reviewer #2: No

Reviewer #3: No

---

## [Author Response · Author response to Decision Letter 0]

20 Apr 2024

Reviewer #1:

1-We believe this article is suitable for publication in the journal although major revisions are needed. The main strengths of this paper are that it addresses an interesting and very timely question and provides a clear answer, with some limitations. Certainly, the authors should better highlight the limitations of the current paper.

Response: We appreciate your pointing out our shortcomings. We have further refined and emphasized the limitations of the current paper, specifically outlined in lines 483-501. Please see the below revisions.

“However, there are several limitations in this study. 1) Only English and Chinese language databases were surveyed, therefore, omissions may exist. All the included studies were sourced from China because of the main location for Chinese medicine research, which may have regional and ethnic influences. Further research is thus required to extend the current findings to other populations and regions. 2) Only 13 studies reported random sequence generation, and none followed the blinding method, which may have introduced potential selection and detection biases. Therefore, the GRADE rating for the evidence quality of these datasets might be low or very low. However, the sensitivity analysis showed that the outcomes were reliable. The data will be constantly updated with the publication of new high-quality studies. 3) Only 8 studies reported the long-term survival in PLC patients, thus, warrants further research. 4) Although all studies reported quality control, only 8 studies included chemical analysis; the active ingredients of TMPs in other studies were unclear and required clarification from the drug manufacturers or research studies, which may cause bias. In addition, sensitivity and sub-cohort analyses were carried out to mitigate the effect of heterogeneity and the resulting bias for specific traditional medicine within TMPs. 5) The majority of the studies reviewed within this investigation did not adhere stringently to reporting standards for CONSORT Extension for Chinese Herbal Medicine Formulas.”

2- A linguistic revision is needed.

Response: Thank you for pointing out this mistake. We have checked and revised the manuscript carefully with the help of experts.

3- The background of the changing scenario of medical treatment in HCC should be better discussed, and some recent papers regarding this topic should be included (PMID: 32772560; PMID: 36633661; PMID: 34798793; PMID: 35031442).

Response: Thank you for your comments. This point you mentioned is very important. We have incorporated recent advancements in the treatment of HCC, particularly focusing on immunotherapy and combination immunotherapy with targeted therapies, in the “Introduction” section. In addition to citing the four articles suggested by you, we have also included recent relevant literature. Please refer to lines 64-77 for specifics. Please see the below revisions.

“In recent years, in addition to targeted therapies, significant breakthroughs in immunotherapy have also been made. A meta-analysis revealed that in patients undergoing immunotherapy (either monotherapy or in combination with other anticancer agents), the pooled odds ratio was 1.67 [95% confidence interval (CI): 1.52 - 1.84]. Compared to control treatments, immune checkpoint inhibitors have indicated substantially increased rates of achieving complete response (CR) (15). Moreover, gender was observed to influence the efficacy of immune checkpoint inhibitors in cancer patients, with males generally experiencing greater overall benefit after this therapy (16, 17). It has been indicated that patients with advanced hepatocellular carcinoma (HCC) can benefit from immunotherapy (18). Furthermore, for treating HCC, many studies have revealed the efficacy of immunotherapy, such as Tremelimumab + Durvalumab and Durvalumab alone (19), as well as the combination of immunotherapy with targeted therapy, such as Atezolizumab + Bevacizumab (20). The National Comprehensive Cancer Network (2024) guidelines have also approved these treatment strategies.”

References

15. Santoni M, Rizzo A, Kucharz J, Mollica V, Rosellini M, Marchetti A, et al. Complete remissions following immunotherapy or immuno-oncology combinations in cancer patients: the MOUSEION-03 meta-analysis. Cancer Immunol Immunother. 2023;72(6):1365-79.

16. Santoni M, Rizzo A, Mollica V, Matrana MR, Rosellini M, Faloppi L, et al. The impact of gender on The efficacy of immune checkpoint inhibitors in cancer patients: The MOUSEION-01 study. Crit Rev Oncol Hematol. 2022;170:103596.

17. Conforti F, Pala L, Bagnardi V, De Pas T, Martinetti M, Viale G, et al. Cancer immunotherapy efficacy and patients' sex: a systematic review and meta-analysis. Lancet Oncol. 2018;19(6):737-46.

18. Rizzo A, Ricci AD, Brandi G. Systemic adjuvant treatment in hepatocellular carcinoma: tempted to do something rather than nothing. Future Oncol. 2020;16(32):2587-9.

19. Abou-Alfa GK, Lau G, Kudo M, Chan SL, Kelley RK, Furuse J, et al. Tremelimumab plus Durvalumab in Unresectable Hepatocellular Carcinoma. NEJM Evid. 2022;1(8):EVIDoa2100070.

20. Cheng AL, Qin S, Ikeda M, Galle PR, Ducreux M, Kim TY, et al. Updated efficacy and safety data from IMbrave150: Atezolizumab plus bevacizumab vs. sorafenib for unresectable hepatocellular carcinoma. J Hepatol. 2022;76(4):862-73.

Reviewer #2:

1- Some typos and grammatical errors ought to be corrected through the whole manuscript.

Response: Thank you for pointing out this mistake. We have checked and revised the manuscript carefully with the help of experts.

2- The article has too many abbreviations, which might cause confusion to readers. I recommend to reduce the number of abbreviated terms within the manuscript.

Response: Thank you for the comments. Your comments make our paper more intuitive. We have reduced the use of abbreviated terms in the re-submitted manuscript. All abbreviated terms are now included in the "Abbreviations" section to ensure clarity for the readers.

3- The introduction suffers from poor discussion about Cancer/VEGFR relationship and VEGFR-TK inhibitors. Please improve.

Response: Thank you for your comments. This point you mentioned is very important. We have incorporated information regarding the Cancer/VEGFR relationship and VEGFR-TK inhibitors into the “Introduction” section in lines 38-52. Please see the below revisions.

“The Vascular Endothelial Growth Factor (VEGF) family comprises VEGF-A, VEGF-B, VEGF-C, VEGF-D, and placental growth factor (2). These proteins predominantly bind VEGF receptor-1 (VEGFR-1) or Fms-like Tyrosine Kinase-1, and VEGFR-2 (also called kinase insert domain-containing receptor) (2, 3). This interaction activates downstream signaling pathways crucial for endothelial cell proliferation, differentiation, and migration, as well as the regulation of vascular permeability, which are essential for angiogenesis. Angiogenesis ensures a steady supply of oxygen and nutrients during tumor development and progression. VEGF, secreted by tumor cells and their microenvironment, binds to VEGFR-2 and exerts a pivotal role in vascular permeability and neo-angiogenesis (4). In 1993, a monoclonal antibody targeting and neutralizing VEGFA was identified to inhibit tumor growth in xenograft models, which opened translational avenues for targeting VEGF-VEGFR signaling (5). These therapeutic agents can be broadly categorized into those which target the VEGF ligand and those which inhibit the cell surface receptor (6). VEGFR-TKIs is a class of small molecules targeted therapies that can selectively inhibit the phosphorylation of tyrosine kinase receptors, thereby suppressing tumor angiogenesis.”

References

2. Dakowicz D, Zajkowska M, Mroczko B. Relationship between VEGF Family Members, Their Receptors and Cell Death in the Neoplastic Transformation of Colorectal Cancer. Int J Mol Sci. 2022;23(6).

3. Ferrara N, Gerber HP, LeCouter J. The biology of VEGF and its receptors. Nat Med. 2003;9(6):669-76.

4. Liu G, Chen T, Ding Z, Wang Y, Wei Y, Wei X. Inhibition of FGF-FGFR and VEGF-VEGFR signalling in cancer treatment. Cell Prolif. 2021;54(4):e13009.

5. Kim KJ, Li B, Winer J, Armanini M, Gillett N, Phillips HS, et al. Inhibition of vascular endothelial growth factor-induced angiogenesis suppresses tumour growth in vivo. Nature. 1993;362(6423):841-4.

6. Ferrara N. Vascular endothelial growth factor: basic science and clinical progress. Endocr Rev. 2004;25(4):581-611.

4- “Meta-analysis was performed through two scientists (HBY and JQH) independently”. I do no prefer to cite the authors’ names here. It is much better to talk about it generally.

Response: Thank you for the suggestion. We have removed the authors' names from the methodology section.

5- It is not recommended to use abbreviations in heading sections (e.g QoL, AFP, ADRs, etc…)

Response: Thank you for your comments. In the resubmitted manuscript, we have utilized full names in the heading sections.

6- The conclusion part has to be improved. It should include deep insights.

Response: We appreciate for pointing out our shortcomings. We have enriched the conclusions with deeper insights in lines 524-539. Please see the below revisions.

“This investigation indicated that TMPs + VEGFR-TKIs co-treatment has enhanced efficacy and reduced risk of ADRs, for middle-advance PLC patients. Furthermore, only the “TMPs + sorafenib” combination improved the one-year survival rate in long-term prognosis. In addition, TMPs + lenvatinib co-treatment has better ORR than the combination of TMPs with other VEGFR-TKIs. When it comes to DCR, TMPs + sorafenib co-treatment was better. Moreover, it was indicated that TMPs can reduce AFP and need further exploration. Based on the meta-analysis, it was recommended that for middle-advanced PLC treatment, the following traditional medicines should be combined with VEGFR-TKIs: P. ginseng, T. robiniophila Murr., S. nigrum, R. glutinosa, A. sinensis, Citrus × aurantium, C. aromatica, P. persica, E. sinensis walker, S. barbata, S. s. mutilans, M. pentadactyla Linnaeus, and G. glabra, as they have a significant effect on ORR/DCR. Additionally, following combinations of traditional medicines with VEGFR-TKIs were also recommended: E. sinensis Walker + S. nigrum + R. glutinosa; C. aromatica + P. ternate + Citrus × aurantium; S. barbata + S. s. mutilans. However, further verification is required with well-designed and high-quality clinical trials adhering to CONSORT Extension for Chinese Herbal Medicine Formulas guidelines.”

7- All figures’ resolution have to be improved.

Response: Thank you for the suggestion. We have repositioned the figures according to the recommended format of PLOS ONE, adjusting the resolution to 300 dpi to meet your requirements. 

Reviewer #3:

1- The rationale for this review and its aims should be more clear for the readers.

Response: We appreciate for pointing out our shortcomings. In the Introduction section, we have reiterated the rationale and aims of this review in lines 97-105. Please see the below revisions.

“Much literature indicated that VEGFR-TKIs have enhanced the efficacy and reduced adverse drug reactions in PLC patients treated with combined TMPs + VEGFR-TKIs therapy compared to VEGFR-TKIs alone. These findings suggest that compared to current therapeutic modalities, TMPs may serve as effective complementary or alternative treatments for PLC with more favorable risk-benefit profiles. However, because of the limited number of clinical trials investigating the co-treatment of TMPs + VEGFR-TKIs and their small sample sizes, the evidence for its potential use is less convincing. Therefore, our objective is to assess the therapeutic efficacy and safety of TMPs in combination with VEGFR-TKIs for middle-to-advanced PLC, aiming to provide substantial evidence.”

2- I wonder if the authors could provide more recent studies from 2023, as I can see there is no any study from 2023.

- Also, it is worth mentioning that there is a limited number of studies from 2022.

Response: Thank you for the suggestion. Due to the extended submission cycle, our initial manuscript had an outdated literature search timeframe. Consequently, we have updated our search to include all relevant literature up to April 12, 2024. After removing duplicates, a total of 158 relevant articles published in 2023 were identified. However, only a few met our inclusion criteria. Through literature screening, three articles passed the abstract screening stage. During full-text screening, one article was excluded due to the use of non-fixed TMPs in the trial, and another was excluded as it was not an RCT. Ultimately, one additional article meeting the criteria was included. In total, our study encompasses 26 articles, including six from 2022 and one from 2023. The relatively low number of articles from 2023 may be attributed to ongoing trials whose results have not yet been published, as well as updates to treatment guidelines. Moving forward, we will continue to monitor developments in this field and update our study accordingly.

3- The manuscript contains some errors and inaccuracies in terms of typography. I kindly request you to review it carefully and make the necessary revisions.

Response: Thank you for pointing out this mistake. We have checked and revised the manuscript carefully according to PLOS ONE's “Submission Guidelines”.

---

## [Decision Letter · Decision Letter 1]

21 Jun 2024

PONE-D-23-16652R1Combinatorial Treatment with Traditional Medicinal Preparations and VEGFR-Tyrosine Kinase Inhibitors for Middle-Advanced Primary Liver Cancer: A Systematic Review and Meta-AnalysisPLOS ONE

Dear Dr. Huibo Yu,

Thank you for submitting your manuscript to PLOS ONE. After careful consideration, we feel that it has merit but does not fully meet PLOS ONE’s publication criteria as it currently stands. Therefore, we invite you to submit a revised version of the manuscript that addresses the points raised during the review process.

We look forward to receiving your revised manuscript.

Kind regards,

Hazem Osama Radwan Elkady

Academic Editor

PLOS ONE

Journal Requirements:

Reviewers' comments:

Reviewer's Responses to Questions

**Comments to the Author**

1. If the authors have adequately addressed your comments raised in a previous round of review and you feel that this manuscript is now acceptable for publication, you may indicate that here to bypass the “Comments to the Author” section, enter your conflict of interest statement in the “Confidential to Editor” section, and submit your "Accept" recommendation.

Reviewer #2: All comments have been addressed

Reviewer #3: (No Response)

Reviewer #4: (No Response)

2. Is the manuscript technically sound, and do the data support the conclusions?

Reviewer #2: Partly

Reviewer #3: (No Response)

Reviewer #4: Partly

3. Has the statistical analysis been performed appropriately and rigorously? 

Reviewer #2: I Don't Know

Reviewer #3: (No Response)

Reviewer #4: No

4. Have the authors made all data underlying the findings in their manuscript fully available?

Reviewer #2: Yes

Reviewer #3: (No Response)

Reviewer #4: Yes

5. Is the manuscript presented in an intelligible fashion and written in standard English?

Reviewer #2: Yes

Reviewer #3: (No Response)

Reviewer #4: No

6. Review Comments to the Author

Reviewer #2: All points raised from my side were addressed by authors. Hence, the paper can be accepted in its current form.

Reviewer #3: (No Response)

Reviewer #4: This manuscript reports meta-analysis results for comparing combinatorial treatment with traditional medicinal preparations and VEGFR-Tyrosine kinase inhibitors for middle to advanced primary liver cancer. The following are my questions/comments.

Line 144, what the time point of the inception for the literature search?

Line 177, what do you mean dichotomous datasets or continuous datasets? Do you mean that the outcomes are dichotomous or continuous variables?

Nowhere to indicate which results are from Tables 3-5 in the manuscript. It needs to cite Tables 3-5 where their results are described in the text.

To be easily understood, for all forest plots, please add “Favors” before each treatment group label, e.g “Favors TMP plus VEGFR-TKIs” and “Favors VEGFR-TKIs”.

For Figures 4-7, the term “Risk Ratio” may be confusing for readers since some of these outcomes (ORR, DCR, or one-year OS) are the bigger the better. The bigger values of “Risk Ratio” don’t mean bad Risk. It’s better to use “Fold Change” to replace “Risk Ratio” and make a note to indicate that Flod Change represents the ratio of the studied outcome (ORR, DCR, one-year OS, and ADR) between the co-treatment group versus the mono treatment.

For Figure 8, please add the title to the horizontal Axis. Please add Egger’s test for Fig. 8 C and D.

7. PLOS authors have the option to publish the peer review history of their article (what does this mean?). If published, this will include your full peer review and any attached files.

Reviewer #2: No

Reviewer #3: No

Reviewer #4: No

---

## [Author Response · Author response to Decision Letter 1]

2 Aug 2024

Response to Editors：

Response: Thank you for your comments. We rechecked the references and confirmed that they are complete and accurate.

Response to Reviewers：

Reviewer #4:

-Line 144, what the time point of the inception for the literature search?

Response: We appreciate your pointing out our shortcomings. We have added the inception time information on lines 143-144. Please see the below revisions.

“All randomized controlled trials, published in both Chinese and English, were searched from January 1, 2000 until April 12, 2024.”

-Line 177, what do you mean dichotomous datasets or continuous datasets? Do you mean that the outcomes are dichotomous1 or continuous variables?

Response: Thank you for the comments. By "dichotomous datasets" and "continuous datasets," we were referring to dichotomous and continuous variables. We have revised the terminology in lines 176-179 to clarify this and eliminate any potential confusion for the readers. Please see the below revisions.

“Using the risk ratio (RR), which represents the ratio of the studied outcomes between the co-treatment group and the mono treatment group, along with 95% confidence intervals (CI), dichotomous variables were identified. For continuous variables, the standardized mean difference (SMD) with 95% CI was employed.”

-Nowhere to indicate which results are from Tables 3-5 in the manuscript. It needs to cite Tables 3-5 where their results are described in the text.

Response: Thank you for pointing out this mistake. In the revised manuscript, we referenced Table 3-4 on lines 284-285 and Table 5 on line 367.

Lines 284-285 “Sub-cohort assessments for ORR and DCR were performed for the target drug regimen, treatment course, KPS score, and TMP form (Table 3 and Table 4).”

Line 367 “Table 5A and Table 5B present the sub-cohort analysis of ORR and DCR for these traditional medicines and their combinations.”

-To be easily understood, for all forest plots, please add “Favors” before each treatment group label, e.g “Favors TMP plus VEGFR-TKIs” and “Favors VEGFR-TKIs”.

Response: Thank you for the comments. We have added "Favors" before each treatment group label in all the forest plots to ensure they are easier for readers to understand.

-For Figures 4-7, the term “Risk Ratio” may be confusing for readers since some of these outcomes (ORR, DCR, or one-year OS) are the bigger the better. The bigger values of “Risk Ratio” don’t mean bad Risk. It’s better to use “Fold Change” to replace “Risk Ratio” and make a note to indicate that Flod Change represents the ratio of the studied outcome (ORR, DCR, one-year OS, and ADR) between the co-treatment group versus the mono treatment.

Response: Thank you very much for your insightful comments. We appreciate your valuable suggestions. In response, we have added an explanation of the risk ratio on lines 183-184. Additionally, we have included a legend in Figure 4 where the RR value first appears to help clarify any potential confusion for the readers. Our study design adheres to the PRISMA guideline, which recommend using the risk ratio to calculate effect sizes. The GRADE criteria for assessing the quality of evidence also requires the use of RR values. After careful consideration and discussion within our team, we kindly request to continue using the term 'Risk ratio' to ensure accuracy and consistency throughout the manuscript. Please see the below revisions.

Lines 183-184 “Using the risk ratio (RR), which represents the ratio of the studied outcomes between the co-treatment group and the mono treatment group,...”

“Fig 4. Forest-plot/pooled RRs comparing the influence of TMPs + VEGFR-TKIs and VEGFR-TKIs on ORR. Risk ratio represents the ratio of the studied outcome between the co-treatment group and the mono treatment group.”

-For Figure 8, please add the title to the horizontal Axis. Please add Egger’s test for Fig. 8 C and D.

Response: Thank you for the comments. We have added the title to the horizontal axis, and we performed Egger’s test for the outcomes of hypertension and hand-foot skin reactions corresponding to Fig. 8 C and D in lines 407-409. Please see the revisions below.

“Moreover, we additionally performed egger’s test on hypertension (p = 0.1206) and hand-foot skin reactions (p = 0.0701), and similarly found no significant publication bias in these outcomes.”

---

## [Decision Letter · Decision Letter 2]

13 Sep 2024

PONE-D-23-16652R2Combinatorial Treatment with Traditional Medicinal Preparations and VEGFR-Tyrosine Kinase Inhibitors for Middle-Advanced Primary Liver Cancer: A Systematic Review and Meta-AnalysisPLOS ONE

Dear Dr. Yu,

Thank you for submitting your manuscript to PLOS ONE. After careful consideration, we feel that it has merit but does not fully meet PLOS ONE’s publication criteria as it currently stands. Therefore, we invite you to submit a revised version of the manuscript that addresses the points raised during the review process.

We look forward to receiving your revised manuscript.

Kind regards,

Hazem Osama Radwan Elkady

Academic Editor

PLOS ONE

Journal Requirements:

Reviewers' comments:

Reviewer's Responses to Questions

**Comments to the Author**

1. If the authors have adequately addressed your comments raised in a previous round of review and you feel that this manuscript is now acceptable for publication, you may indicate that here to bypass the “Comments to the Author” section, enter your conflict of interest statement in the “Confidential to Editor” section, and submit your "Accept" recommendation.

Reviewer #4: (No Response)

2. Is the manuscript technically sound, and do the data support the conclusions?

Reviewer #4: (No Response)

3. Has the statistical analysis been performed appropriately and rigorously? 

Reviewer #4: (No Response)

4. Have the authors made all data underlying the findings in their manuscript fully available?

Reviewer #4: (No Response)

5. Is the manuscript presented in an intelligible fashion and written in standard English?

Reviewer #4: (No Response)

6. Review Comments to the Author

Reviewer #4: The significant level for publication bias is recommended p<0.1 for both Begg and egger [reference]. Please also report the intercepts estimated from Egger’s regressions. To be conservative, authors may add discussions about the potential publication bias among the studied outcomes.

reference: Hayashino Y, Noguchi Y, Fukui T. Systematic evaluation and comparison of statistical tests for publication bias. J Epidemiol. 2005 Nov;15(6):235-43. doi: 10.2188/jea.15.235. PMID: 16276033; PMCID: PMC7904376.

Please revise the table column labels in fig 7, change “M-H, Fixed” to “M-H, Fixed/Random”, so it reflects some of the summarized results are based on random effect models.

7. PLOS authors have the option to publish the peer review history of their article (what does this mean?). If published, this will include your full peer review and any attached files.

Reviewer #4: No

---

## [Author Response · Author response to Decision Letter 2]

20 Sep 2024

Response to Reviewer：

Reviewer #4:

-The significant level for publication bias is recommended p<0.1 for both Begg and egger [reference]. Please also report the intercepts estimated from Egger’s regressions.

Response: We appreciate your insightful suggestion. In response, we have now reported the intercepts estimated from Egger’s regression on lines 384-389. Including these intercepts ensures a more rigorous assessment of potential publication bias in our analysis. Below is the relevant revision:

“Following the results of the funnel plot and Begg’s test, we further conducted Egger’s tests for hypertension (p = 0.1405, intercept = 0.0526) and hand-foot skin reactions (p = 0.0952, intercept = 0.0180) to enhance the rigor of our assessment. A significance level of p < 0.1 is recommended for detecting publication bias in both Begg’s and Egger’s tests [reference]. After a comprehensive evaluation, we concluded that there is potential publication bias in the results for DCR, hypertension, and hand-foot skin reactions.”

[reference]: Hayashino Y, Noguchi Y, Fukui T. Systematic evaluation and comparison of statistical tests for publication bias. J Epidemiol. 2005 Nov;15(6):235-43. doi: 10.2188/jea.15.235. PMID: 16276033; PMCID: PMC7904376.

We believe these changes enhance the robustness of our analysis, and we welcome any further comments you may have.

-To be conservative, authors may add discussions about the potential publication bias among the studied outcomes.

Response: Thank you for your insightful comment. We have added discussions on the potential publication bias across the studied outcomes, as suggested, on lines 515–525. Please see the revisions below:

“Using funnel plots, Begg’s test, and Egger’s test, we identified potential publication bias in DCR, hypertension, and hand-foot skin reactions. This may be attributable to publication bias favoring positive results, wherein studies with significant findings are more likely to be published. Additionally, some of the included studies had small sample sizes, which, if they happened to demonstrate larger effects, may have been more readily published, even though these effects may not be broadly representative. This can lead to potential publication bias. Furthermore, the literature search was restricted to studies published in English and Chinese, which may have contributed to bias as well. We recommend that future research efforts include pre-registered studies and encourage both researchers and journals to publish studies with non-significant findings to mitigate the impact of publication bias.”

-Please revise the table column labels in fig 7, change “M-H, Fixed” to “M-H, Fixed/Random”, so it reflects some of the summarized results are based on random effect models.

Response: Thank you for pointing out this mistake. we revised the labels in Fig7.

---

## [Decision Letter · Decision Letter 3]

29 Sep 2024

PONE-D-23-16652R3Combinatorial Treatment with Traditional Medicinal Preparations and VEGFR-Tyrosine Kinase Inhibitors for Middle-Advanced Primary Liver Cancer: A Systematic Review and Meta-AnalysisPLOS ONE

Dear Dr.  Yu,

Thank you for submitting your manuscript to PLOS ONE. After careful consideration, we feel that it has merit but does not fully meet PLOS ONE’s publication criteria as it currently stands. Therefore, we invite you to submit a revised version of the manuscript that addresses the points raised during the review process.

We look forward to receiving your revised manuscript.

Kind regards,

Hazem Osama Radwan Elkady

Academic Editor

PLOS ONE

Journal Requirements:

Reviewers' comments:

Reviewer's Responses to Questions

**Comments to the Author**

1. If the authors have adequately addressed your comments raised in a previous round of review and you feel that this manuscript is now acceptable for publication, you may indicate that here to bypass the “Comments to the Author” section, enter your conflict of interest statement in the “Confidential to Editor” section, and submit your "Accept" recommendation.

Reviewer #4: (No Response)

2. Is the manuscript technically sound, and do the data support the conclusions?

Reviewer #4: (No Response)

3. Has the statistical analysis been performed appropriately and rigorously? 

Reviewer #4: (No Response)

4. Have the authors made all data underlying the findings in their manuscript fully available?

Reviewer #4: (No Response)

5. Is the manuscript presented in an intelligible fashion and written in standard English?

Reviewer #4: (No Response)

6. Review Comments to the Author

Reviewer #4: Lines 382-385, insignificant publication bias should not include either DCR (p = 0.0852) or hypertension (p = 0.0546) because a significance level of p < 0.1 is recommended for detecting publication bias in both Begg’s and Egger’s tests.

7. PLOS authors have the option to publish the peer review history of their article (what does this mean?). If published, this will include your full peer review and any attached files.

Reviewer #4: No

---

## [Author Response · Author response to Decision Letter 3]

2 Oct 2024

Response to Reviewer：

Reviewer #4:

-Lines 382-385, insignificant publication bias should not include either DCR (p = 0.0852) or hypertension (p = 0.0546) because a significance level of p < 0.1 is recommended for detecting publication bias in both Begg’s and Egger’s tests.

Response: Thank you for pointing out this mistake. We agree that a significance level of p < 0.1 is recommended for detecting publication bias in Begg’s and Egger’s tests. We have revised the section accordingly. Please see the revisions below:

Line 381-384

“These data were validated by Begg’s test, which indicated insignificant publication bias for ORR (p = 0.3323), hand-foot skin reactions (p = 0.3506), and gastrointestinal reactions (p = 0.9562). However, potential publication bias was indicated for DCR (p = 0.0852), and hypertension (p = 0.0546).”

---

## [Editor Report · Decision Letter 4]

24 Oct 2024

Combinatorial Treatment with Traditional Medicinal Preparations and VEGFR-Tyrosine Kinase Inhibitors for Middle-Advanced Primary Liver Cancer: A Systematic Review and Meta-Analysis

PONE-D-23-16652R4

Dear Dr. Huibo Yu,

We’re pleased to inform you that your manuscript has been judged scientifically suitable for publication and will be formally accepted for publication once it meets all outstanding technical requirements.

Kind regards,

Hazem Osama Radwan Elkady

Academic Editor

PLOS ONE
---

## [Editor Report · Acceptance letter]

13 Nov 2024

PONE-D-23-16652R4 

PLOS ONE

Dear Dr. Yu, 

I'm pleased to inform you that your manuscript has been deemed suitable for publication in PLOS ONE. Congratulations! Your manuscript is now being handed over to our production team.

Kind regards, 

on behalf of

Dr. Hazem Osama Radwan Elkady 

Academic Editor

PLOS ONE